# The Par3 polarity protein is an exocyst receptor essential for mammary cell survival

Syed Mukhtar Ahmed[1] & Ian G. Macara[1]

The exocyst is an essential component of the secretory pathway required for delivery of basolateral proteins to the plasma membranes of epithelial cells. Delivery occurs adjacent to tight junctions (TJ), suggesting that it recognizes a receptor at this location. However, no such receptor has been identified. The Par3 polarity protein associates with TJs but has no known function in membrane traffic. We now show that, unexpectedly, Par3 is essential for mammary cell survival. Par3 silencing causes apoptosis, triggered by phosphoinositide trisphosphate depletion and decreased Akt phosphorylation, resulting from failure of the exocyst to deliver basolateral proteins to the cortex. A small region of PAR3 binds directly to Exo70 and is sufficient for exocyst docking, membrane-protein delivery and cell survival. PAR3 lacking this domain can associate with the cortex but cannot support exocyst function. We conclude that Par3 is the long-sought exocyst receptor required for targeted membrane-protein delivery.

[1] Department of Cell and Developmental Biology, Vanderbilt University School of Medicine, Nashville, Tennessee 37232, USA. Correspondence and requests for materials should be addressed to I.G.M. (email: ian.g.macara@vanderbilt.edu).

Cell polarity is defined largely by the segregation of the cell cortex into domains populated by functionally distinct membrane proteins. Throughout the animal kingdom, these domains are formed by a conserved group of polarity proteins, which includes Par3, Par6 and atypical protein kinase C (aPKC)[1]. Mammalian epithelia are segregated into apical and basolateral domains separated by tight junctions (TJs)[1,2]. Par3, a large scaffold protein at the apex of the polarity signalling network, localizes to TJs and to the lateral membrane just beneath them[3,4]. The biological functions of Par3 are not fully understood.

It interacts with Par6 and aPKC and is necessary to recruit aPKC to the apical cortex, which mediates the exclusion of basolateral proteins from the apical domain[3,5–8]. It also sequesters TIAM1, a Rac exchange factor, so as to spatially restrict the production of RacGTP[8], and has been reported to bind many other proteins including the phosphoinositide phosphatase Pten and the exocyst complex[9–13]. However, the biological meaning of these interactions remains mostly obscure. In addition to the regulation of cell polarity, Par3 is in some tissues required for cell survival. Silencing of Par3 expression in the mammary gland, for

**Figure 1 | Loss of Par3 induces apoptosis.** (**a,b**) Knockdown of Par3 in NMuMG or Eph4 cells induces cleaved Caspase-3 activation. The experiments were repeated at least three times and representative blots are shown. (**c**) Knockdown of Par3 in NMuMG cells induces Parp cleavage. (**d**) Immunofluorescence staining of NMuMG cells showing the activity of Caspases 3 and 7 in NMuMG cells using CellEvent Caspase-3/7 Green ReadyProbes (green) and phalloidin Alexa594 (red) and Hoechst 33342 (blue). Scale bars, 50 μm. (**e**) Quantification of cleaved Caspase-3 staining in NMuMG cells. Data represented as the mean percentage of cleaved Caspase-3 compared with control, the graph shows mean ± s.e.m. for four experimental replicates and P value was calculated by the Student's t-test. (**f**) Phase contrast images of NMuMG cells treated with shLuc, shPar3 or shPar3 + YFP-hPAR3b. Scale bars, 200 μm. (**g**) Immunoblot showing cleaved Caspase-3 induction by Par3 depletion can be reversed by expression of YFP-hPAR3b. (**h**) Co-culture of NMuMG cells expressing shLuc plus GFP (green) or shPar3 plus RFP (red) and stained with Hoechst 33342 (blue). The experiment was repeated three times and a representative image is shown. The boxed region is shown as magnified below. Scale bar, 100 μm. (**i**) Quantification of cells with normal or condensed nuclear morphology in the experiment shown in **h**. Error bars represent mean ± s.e.m.

instance, strongly enhances apoptosis, both *in vivo* and in primary mammosphere cultures[7,14]. Deletion of Par3 in the mouse epidermis also promotes apoptosis[15], but the underlying mechanism is unknown.

Steady-state levels of membrane proteins depend not only on transcription/translation but also on the rates of exocytosis and endocytosis, both subject to multiple levels of control[16–18]. Delivery of cargo to basolateral membranes requires the exocyst, discovered in budding yeast and conserved throughout the eukaryotes[17,19,20]. The exocyst is a complex of eight subunits, that tethers vesicles to the plasma membrane (PM) through interactions with SNARES, small GTPases and accessory proteins[17,21–24]. In mammalian epithelia, the exocyst can associate generally with membranes through the interaction of Exo70 (*Exoc4*) and Sec3 (*Exoc1*) subunits with phosphoinositides[25–29], but this interaction cannot provide the targeting information needed to deliver the exocyst and membrane proteins to the TJ zone. Such localized docking in epithelia suggests that the exocyst must recognize a receptor in this region of the membrane. However, no such receptor has been identified.

We now demonstrate that Par3 is essential for the survival of mammary epithelial cells (MECs) through recruitment of exocyst to intercellular junctions. Silencing of Par3 causes a substantial drop in phosphoinositide-(3,4,5)-trisphosphate (PIP$_3$) levels, which depresses AKT phosphorylation, thereby triggering apoptosis. This cascade of effects is caused by a defect in lateral membrane-protein delivery, and loss of exocyst localization to TJs. Silencing of exocyst subunits phenocopies Par3 depletion. We identify a small region of PAR3 that binds the exocyst and this isolated domain is sufficient, if targeted to membranes, to rescue cell survival in the absence of endogenous Par3. A full-length mutant of Par3 that lacks this domain cannot rescue survival. Therefore, we conclude that Par3 functions as an exocyst receptor, enabling polarized recruitment to the TJ zone of the epithelial PM.

## Results

**Par3 depletion in mammary epithelial cells induces apoptosis.** To gain insight into why Par3 is required for cell survival, we used two polarized epithelial cell lines, NMuMG and Eph4, which are derived from the murine mammary gland. Both cell lines have properties similar to luminal epithelial cells. Par3 properly localizes with the TJ protein Zo-1 and is distinctly separated from β-catenin, which marks the lateral membrane (Supplementary Fig. 1a–h). We silenced Par3 with a previously validated short hairpin (shPar3)[7,30]. Within 2 days, cleaved Caspase-3 (Casp3) and cleaved PARP were detected in NMuMG cells (Fig. 1a,c). By 3 days, the majority of the cells had detached from the dish, and many of the remaining cells (~60%) stained positive for cleaved Casp3, as compared with cells transduced with a control shRNA, shLuc (Fig. 1d,e and Supplementary Fig. 2a,b). Flow cytometry using Annexin V (Anxa5) exposure to mark apoptosis also revealed extensive cell death (~55% of the population) in response to Par3 depletion (Supplementary Fig. 2c). Silencing of Par3 in Eph4 mammary cells as well as primary mammary epithelial cells also induced Casp3 cleavage (Fig. 1b and Supplementary Fig. 2d). Expression of human PAR3 fused to YFP, which is insensitive to the mouse shRNA, completely rescued the survival of cells depleted of endogenous Par3 (Fig. 1f,g).

**Apoptosis caused by loss of Par3 is cell autonomous.** We next asked if apoptosis triggered by depletion of Par3 is cell autonomous. NMuMG cells were transduced separately with lentivirus to express shLuc plus YFP, or shPar3 plus RFP, then mixed at a

1:1 ratio and co-cultured. Three days post transduction, when apoptosis had begun, we stained with Hoechst 33342 to identify cells with condensed nuclei. Notably, whereas green cells expressing shLuc were phenotypically normal, neighbouring red cells expressing shPar3 exhibited condensed nuclei and extrusion from the epithelial sheet (yellow arrowheads) (Fig. 1h). About 60% of the cells expressing shPar3 showed condensed nuclear staining, whereas <10% of the control cells showed a similar phenotype (Fig. 1i). We conclude that apoptosis induced by Par3 depletion is cell autonomous.

**Apoptosis results from the inactivation of Akt.** Depletion of Par3 in NMuMG cells resulted in decreased phosphorylation of Foxo3a at residue S253 (Fig. 2a). Knockdown of Par3 also decreased the phosphorylation of Bad at S136 (Fig. 2b). The pro-apoptotic protein Bim showed a concomitant increase in expression level (Fig. 2c).

Foxo3a and Bad are known targets of Akt[31], suggesting that loss of Par3 might somehow interfere with Akt activation. Indeed, silencing of Par3 caused a fivefold decline in pAkt (Fig. 2d,e). Immunofluorescence showed a similar decrease in pAkt (Supplementary Fig. 4a). Similarly, Par3 depletion from Eph4 also reduced pAkt levels (Supplementary Fig. 4e). To assess if a similar phenotype occurs in primary MECs we purified luminal mammary epithelial cells from mice by fluorescence-activated cell sorting (FACS; Fig. 2f and Supplementary Fig. 3a,b). Depletion of Par3 in these freshly isolated luminal cells also induced cell death and extrusion within 3 days post transduction with shPar3 (Fig. 2g). Phospho-Akt was decreased and cleaved Casp3 increased (Fig. 2h). We note that the slight deformation of bands seen in the shPar3 treated samples is caused by the manner cells had to be harvested to minimize cell losses, which inadvertently left some BSA from the culture media, resulting in distortion in gel migration. Although the phenotype was less severe compared with NMuMG cells, the effects were significant (Fig. 2i), demonstrating that the effects of Par3 depletion on survival are intrinsic to mammary epithelial cells and are not an artefact of using cell lines.

Akt is phosphorylated by Pdk1, which is activated by PIP$_3$ (refs 32,33). Consistent with this mechanism inducing apoptosis, treatment of NMuMG cells with the PI3-K inhibitor LY294002 triggered a threefold increase in AnnexinV-positive cells (Supplementary Fig. 4c,d). Moreover, 50 nM of the Akt inhibitor MK-2206 induced cell death within 20 h of drug treatment (Supplementary Fig. 4b). Finally, we found that a constitutively active mutant of human AKT1 (AKT-CA)[34] efficiently prevented the induction of Casp3 cleavage in cells depleted of Par3 (compare lanes 2 and 3, Fig. 2j). Together, these data show that mammary epithelial cells are exquisitely sensitive to survival signals from Akt, and that, unexpectedly, Par3 expression is essential to maintain this signal.

**Depletion of Par3 reduces PIP$_3$ levels.** Par3 might act by supporting PIP$_3$ production at the PM, or by more directly regulating Akt phosphorylation. To distinguish these possibilities, we quantified PIP$_3$ in NMuMG cells by immunoassay. The level was decreased by 50% in cells depleted of Par3 (Supplementary Fig. 4f). We also used a biosensor, consisting of a PH domain-AKT-GFP fusion, to localize PIP$_3$ in intact cells, together with mApple to mark transfected cells (Fig. 2k). Expression of shPar3 caused a twofold reduction in PH-AKT-GFP localization to the cell cortex (Fig. 2l).

**pAkt is not maintained through Pten association with Par3.** PIP$_3$ is hydrolysed by the tumour suppressor phosphatase and

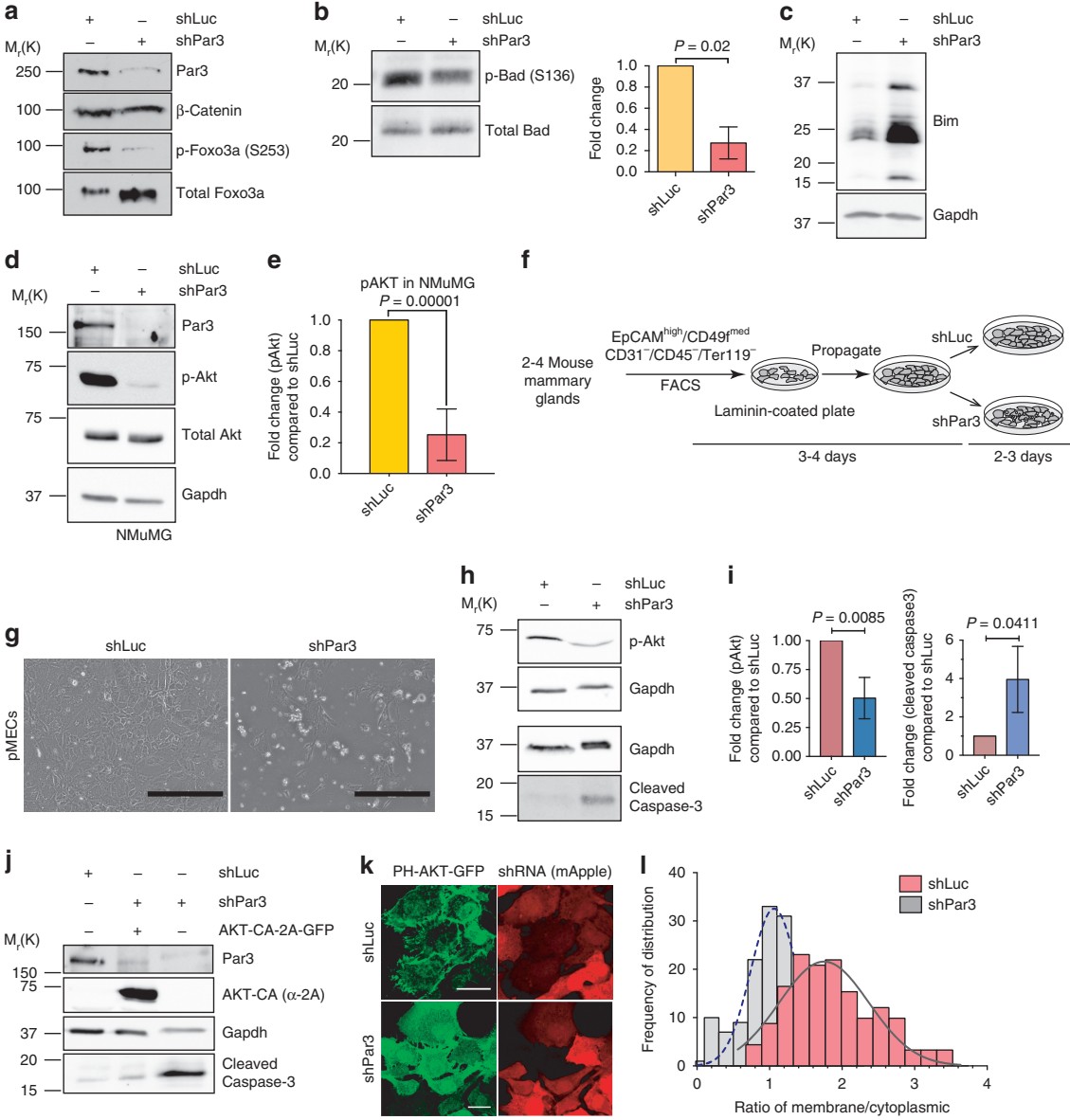

**Figure 2 | Loss of Par3 reduces pAkt and PtdIns-(3,4,5)-P₃ levels.** Immunoblots showing changes in (**a**) phospho-Foxo3a; p-Foxo3a (S253), (**b**) phospho-Bad(S136); p-Bad, mean ± s.d., $n = 3$. (**c**) Bim, and (**d**) phospho-Akt (p-Akt) in NMuMG cells. (**e**) Quantification of p-Akt levels in NMuMG cells, mean ± s.d., $n = 4$. (**f**) Schematic representation of primary murine mammary luminal epithelial cell isolation and culture, also see Supplementary Fig. 3; details described in methods. (**g**) Phase contrast images of primary MECs (pMECs), 3 days after transduction with shLuc or shPar3. Scale bars, 400 μm. (**h**) p-Akt and cleaved Caspase-3 levels in pMECs upon treatment of cells with shLuc or shPar3. (**i**) Quantification of p-Akt and cleaved Caspase-3 in pMECs from three independent experiments. Error bars, mean ± s.d. (**j**) Immunoblot analysis showing cleaved Caspase-3 levels upon shLuc or shPar3 expression in the presence or absence of constitutively active AKT1(T308D, S473D) (AKT-CA). (**k**) Localization of PH-AKT-GFP in the presence of shLuc or shPar3. mApple was used as an internal marker for hairpin vector transduction. Scale bars, 20 μm. (**l**) Frequency distribution histogram showing membrane to cytoplasmic ratio of PH-AKT-GFP localization. P values for all statistics calculated using Student's t-test.

tensin homology protein (Pten), which has been previously shown to bind Par3 both in *Drosophila* and mammalian cells[10,11,35]. Therefore, we asked whether Pten regulates Akt in NMuMG cells. As expected, silencing of Pten increased Akt phosphorylation. Co-expression of shPten also reversed the drop in Akt phosphorylation and the Casp3 cleavage caused by shPar3 (Fig. 3a, compare lanes 2–4). Next, we asked if the interaction between Pten and Par3 is important for cell survival. We first attempted to confirm the interaction by co-immunoprecipitation with either endogenous Pten or over-expressed HA-PTEN, but could not detect significant binding, under conditions in which endogenous aPKC co-precipitated robustly with Par3 (Supplementary Fig. 4g). Nonetheless, based

on data from synthetic peptide interactions of Pten with Par3 (ref. 10), we mutated the PAR3 PDZ3 domain at two residues reported to be essential for Par3-Pten binding, (R596D,K598D)[10]. This mutant, PAR3(R596D,K598D), efficiently rescued cell survival (Fig. 3b). Together, these experiments argue that a Par3-Pten interaction is not involved in mammary cell survival signalling.

**Post-Golgi membrane trafficking is perturbed by Par3 loss.** An alternative explanation for decreased PIP₃ production would be a failure of PI3-K recruitment to the PM. In mammalian epithelia, PI3-K is associated with the E-cadherin:β-catenin complex[36]. Therefore, we assessed E-cadherin (CDH1) localization in

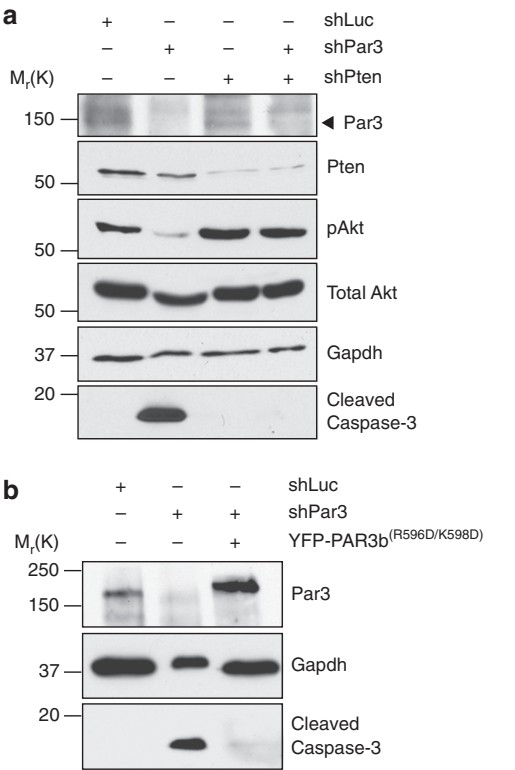

**Figure 3 | Decrease in pAkt upon Par3 knockdown is independent of the interaction between Par3 and Pten.** (**a**) Immunoblot showing cleaved Caspase-3 and pAkt levels in cells treated with shLuc or shPar3 alone or together with shPten. (**b**) Cleaved Caspase-3 levels after expression of GFP-hPAR3(R596D/K598D).

NMuMG and Eph4 cells after Par3 depletion. To block apoptosis we added Caspase inhibitor Ac-DEVD-CHO. Interestingly, silencing of Par3 in both cell types prevented E-cadherin localization to intercellular junctions. Instead, E-cadherin appeared to be trapped in vesicle-like structures (Fig. 4a). Failure of E-cadherin to localize properly at the cell cortex did not significantly affect its stability (Fig. 4b,c). To evaluate specificity, we stained for another lateral membrane protein, $Na^+$-$K^+$-ATPase (NKA; Fig. 4a). Strikingly, Par3 depletion in NMuMG or Eph4 cells drastically impaired NKA localization to the PM, and the protein instead accumulated in an intracellular compartment, in a similar fashion to E-cadherin. The apical membrane protein Muc1 was not mislocalized in response to Par3 silencing (Supplementary Fig. 5a), suggesting that the defect may be restricted to lateral membrane traffic.

An important question was whether this defect is upstream or downstream of the decreased $PIP_3$ levels and Akt activity. Co-depletion of Pten with Par3, which should increase $PIP_3$, did not rescue normal E-cadherin localization (Supplementary Fig. 5b). Furthermore, it is unlikely that the phenotype is caused by disruption of Par3-Pten binding, as expression of PAR3(R596D, K598D) was able to rescue E-cadherin to the PM (Supplementary Fig. 5c). We next treated NMuMG cells with the Akt inhibitor MK2206 (50 μM). Within 8 h of MK2206 treatment, cells began to round up, at which point the cells were fixed and stained for E-cadherin. Despite efficient blocking of Akt phosphorylation, E-cadherin localization at the junctions remained unaffected (Supplementary Fig. 5d). These data support the conclusion that the decreases in $PIP_3$ and in Akt phosphorylation caused by Par3 silencing are consequences rather than causes of the defect in lateral membrane-protein localization.

This failure of lateral membrane proteins to localize at the cortex might be caused by defective delivery or by increased endocytosis[37]. To distinguish these possibilities we first determined whether Par3 depletion impacts temperature-sensitive VSVG-GFP fusion protein exit from the Golgi and sorting to the lateral membrane[38,39]. Cells expressing VSVG-GFP were initially grown at 40 °C to trap the protein in the endoplasmic reticulum. They were then switched to 32 °C for 2.5 h. Control cells, transfected with shLuc shRNA, showed cortical GFP fluorescence. However, Par3 silencing caused the accumulation of GFP-positive spots throughout the cytoplasm, indicative of failed PM delivery of VSVG-GFP (Fig. 4d). We next blocked post-Golgi transport using Brefeldin A (BFA)[40,41]. Giantin staining in NMuMG and Eph4 cells confirmed disruption of the Golgi apparatus. Within 3 h of BFA addition for NMuMG cells, and 7 h for Eph4 cells, E-cadherin was reduced at sites of cell–cell junctions (Fig. 4e). Importantly, treatment of NMuMG cells with BFA for 16 h also led to increased cleaved-Casp3 and was accompanied by a substantial reduction in pAkt (Fig. 4f). Together, these data support a model in which Par3 seems to be required for the delivery of lateral membrane proteins from the Golgi, and failure to deliver these proteins results in the loss of $PIP_3$, decreased Akt activity, and apoptosis.

**E-cadherin co-localizes with Rab11 in Par3-depleted cells.** Rab proteins, particularly Rab11 and Rab8, have been previously shown to coordinate with the exocyst to regulate the asymmetric distribution of proteins in epithelial cells to form polarized tissue structures[13]. To this end, we asked if the intracellular E-cadherin in cells depleted of Par3 co-localizes with Rab11 or Rab8. Because antibodies to these proteins are unreliable, we generated stable NMuMG and Eph4 cell lines expressing YFP-tagged Rab11 or Rab8. In both shLuc- and shPar3-treated cells many of the puncta containing E-cadherin were positive for Rab11 and to a lesser extent Rab8 (Fig. 5a,b). Strikingly, the larger intracellular E-cadherin structures that we saw in Eph4 cells were frequently also Rab11-positive as well as Rab8-positive. From these experiments we conclude that the intracellular E-cadherin puncta are most likely exocytic and/or recycling endocytic vesicles.

**Loss of exocyst proteins phenocopies loss of Par3.** The exocyst is required to capture and guide secretory vesicles to the PM before membrane fusion can occur[42,43]. In mammalian epithelial cells the exocyst complex is required for the delivery of lateral membrane proteins, such as E-cadherin and NKA[44]. The exocyst has been also suggested to converge with the Par3 complex at the apical membrane initiation site during polarization[13], but no requirement of the exocyst for apical membrane delivery has been demonstrated. Moreover, several papers have reported an interaction of exocyst with Par3 or with Par6 or aPKC[13,45,46], but the function of these interactions is obscure. Therefore, we asked if the exocyst is involved in the membrane delivery defect caused by silencing of Par3.

To test if Par3 depletion affects exocyst localization, NMuMG cells were transduced with shLuc or shPar3 and stained for Sec8. Whereas a fraction of Sec8 localized to the cellular junctions in shLuc treated cells, Sec8 was mostly in vesicles and the nuclear compartment in cells depleted of Par3 (Fig. 6a).

Silencing of Sec8 caused E-cadherin mislocalization in both NMuMG and Eph4 cells (Fig. 6b). Moreover, apoptosis was triggered in NMuMG cells by two different shRNAs against Sec8 (Fig. 6c and Supplementary Fig. 6a,b) or an shRNA-targeting Sec10 (EXOC5) (Fig. 6d and Supplementary Fig. 6c), which efficiently reduced the respective protein levels. pAkt levels were also significantly reduced (Fig. 6c,d), and expression of AKT-CA

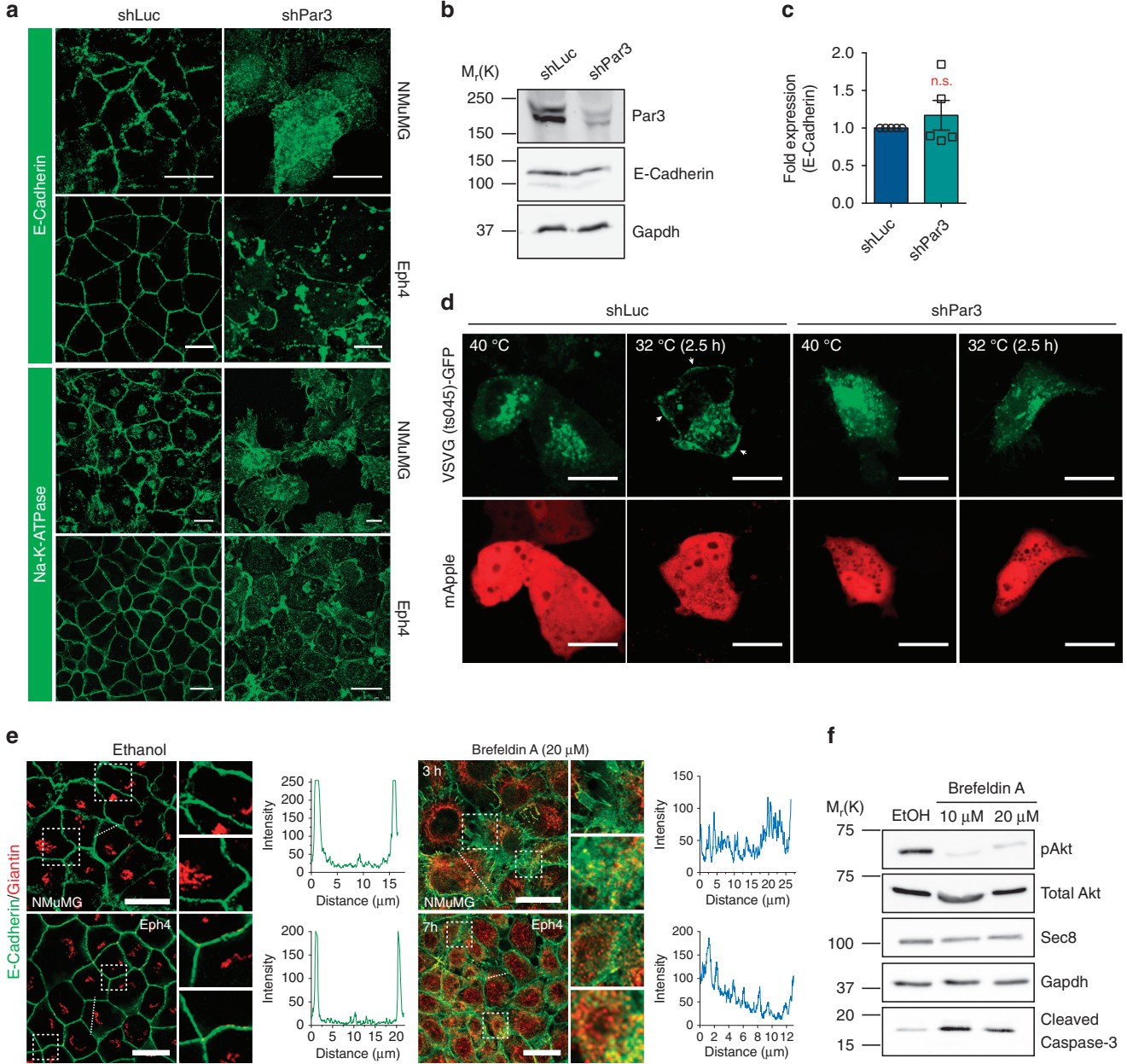

**Figure 4 | Depletion of Par3 in mammary cells affects post-Golgi transport of lateral membrane proteins.** (**a**) E-cadherin and Na$^+$, K$^+$-ATPase localizations in NMuMG and Eph4 cells. Experiments were successfully replicated 3× and representative images are shown. (**b**) Representative immunoblot of E-cadherin expression levels in NMuMG cells treated with shLuc or shPar3 cells. (**c**) Quantification of E-cadherin expression as fold change over control (shLuc); the graph shows mean ± s.e.m.; and statistical significance was assessed from five independent experiments using the Student's *t*-test. (**d**) Localization of temperature-sensitive VSVG mutant VSVG(ts045)-GFP at 40 °C or at 32 °C upon temperature shift in NMuMG cells. Representative images are shown from one experiment. (**e**) Localization of E-cadherin and Giantin upon BFA (20 μM) treatment in NMuMG cells (3 h) or Eph4 cells (7 h). Experiments were replicated three times and representative images are shown. White dashed boxed ROIs shown in enlarged images. White dashed lines shown as fluorescent intensity profiles of E-cadherin staining across a single cell. (**f**) pAkt and cleaved Caspase-3 levels in NMuMG cells after BFA (10 or 20 μM) treatments for 26 h. All scale bars, 20 μm.

in NMuMG cells rescued shSec8-mediated apoptosis (compare lanes 3 and 4, Fig. 6e).

**Exocyst binds a region within the Par3 lysine-rich domain**. Immunoprecipitation with anti-Myc antibody to capture expressed Myc-PAR3 co-precipitated endogenous SEC8 from HEK293T cells (Fig. 7a). Conversely, immunoprecipitation of endogenous Sec8 from NMuMG cells co-precipitated endogenous Par3 (Fig. 7b). Staining for the exocyst component Sec6 in

NMuMG cells expressing GFP-Par3 showed that the two proteins mostly co-localize at the cell cortex. Some intracellular punctate structures were also dual positive for both the proteins (Fig. 7c).

Previously, it was reported that the exocyst can interact with atypical PKCs[46]. Therefore, we tested if the exocyst interaction with Par3 is bridged by aPKC. Depletion of Par3 reduced aPKC co-precipitation with Sec8 (Supplementary Fig. 7a); conversely, however, silencing of aPKC protein levels did not affect Par3 co-precipitation with Sec8 (Supplementary Fig. 7b).

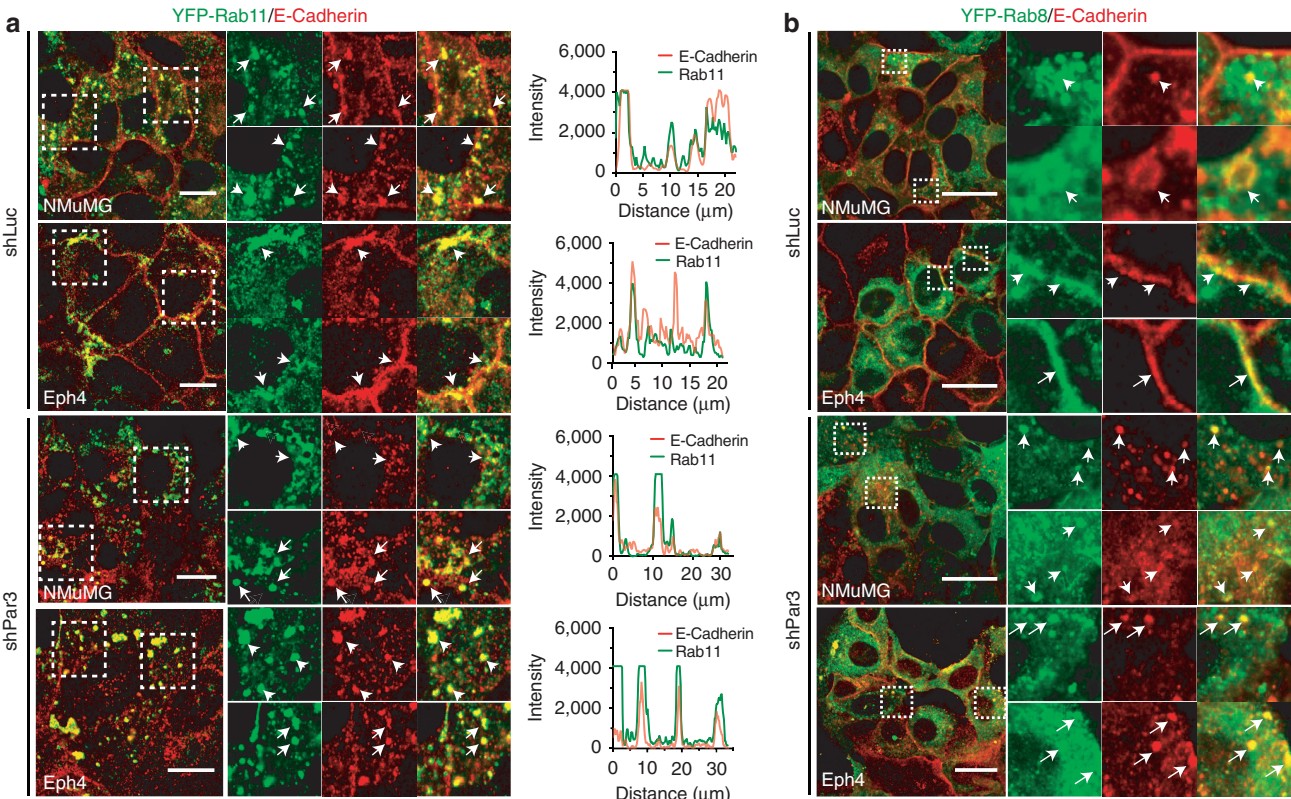

**Figure 5 | E-cadherin co-localizes with Rab11-positive vesicles.** NMuMG and Eph4 cells expressing (**a**) YFP-Rab11, or (**b**) YFP-Rab8 were transduced with shLuc or shPar3, fixed and stained for E-cadherin. Fluorescent intensity profiles taken over a 20–30 µm straight line for both E-cadherin and YFP-Rab11 and overlaid. Confocal images were taken sequentially using 488 and 641 nm excitation lasers. White dashed boxed ROIs are magnified and arrows direct to objects where structures in both channels overlap. Scale bars, 20 µm.

One of the key responses to Par3 depletion is the mislocalization and activation of aPKC[6,7,47]. Hence, we reasoned that cells might undergo apoptosis because of inappropriate substrate phosphorylation by aPKC. However, co-silencing aPKC in cells where Par3 was depleted did not rescue survival (Supplementary Fig. 7c). We conclude that apoptosis in these cells is aPKC-independent and that aPKC is not required for the Par3-exocyst interaction.

Using truncation mutants of Par3, we found that a region containing residues 967-1089, which contains a lysine-rich domain (LRD), was crucial for interaction with the exocyst (Supplementary Fig. 7d–g). This region is highly conserved between mammals (Fig. 7d) and to a lesser extent with the *Drosophila* homologue, Bazooka and *Caenorhabditis elegans* Par3. We added an N-terminal GST tag to the LRD, PAR(967–1045), and purified the protein from *Escherichia coli* (Fig. 7e,f). This GST fusion was incubated with whole-cell lysates from 293 T cells and captured on Glutathione beads. SEC8 robustly co-purified with the GST fusion, whereas little or no SEC8 bound to GST alone (Fig. 7k). Moreover, the purified GST-PAR3 fragment specifically co-precipitated recombinant EXO70-His (Fig. 7m), demonstrating that the interaction with this exocyst component is direct.

The Par3-LRD has been previously shown to associate with phosphoinositides[48]. We confirmed these findings by performing lipid strip and liposome-binding assays using the purified LRD fragment, which showed that this fragment can bind diverse phosphatidylinositol phosphates as well as phosphatidic acid, but had little or no binding to other phospholipids (Fig. 7g–j). We also directly visualized the binding of Par3-LRD to lipids using giant unilamellar vesicles (GUVs) containing phosphatidylinositol phosphates and phosphatidic acid. Purified GST-mApple-Par3-LRD decorated the GUVs within 5–10 min of incubation and was uniformly spread over the GUV surface within 15 min (Fig. 7h).

Next, we asked if binding to phospholipids is essential for the Par3-exocyst interaction. We removed lipids from both purified LRD and HEK293T cell lysates using lipid-adsorbent resin and performed GST capture experiments. Both lipid-clarified and naive samples bound to SEC8 equally well (Fig. 7k). To determine if the multiple Lys residues in the region are necessary for exocyst binding, we treated purified PAR3-LRD with citraconic anhydride at pH 8.0 to block reactive primary amine groups. An O-phthaldialdehyde fluorimetric assay confirmed that at least half of the available Lysines were blocked by citraconic anhydride treatment (Supplementary Fig. 7i). This treatment reduced PAR3-LRD binding to phospholipids, but PAR3-LRD binding to exocyst was unaffected (Fig. 7l and Supplementary Fig. 7h).

Together, these data argue that the exocyst binds to the LRD region of Par3 independent of its phospholipid binding, and probably does not require multiple lysine residues.

**The LRD restores cell survival and E-cadherin localization**. To test if the Par3-exocyst interaction is sufficient to rescue cell survival, we incorporated a myristoylation sequence at the N-terminus of mApple-PAR3(710-1089). This construct efficiently localized at the cell cortex (Fig. 8a). We then expressed shPar3 in wild-type NMuMG cells or in cells stably expressing the myr-mApple-Par3(710-1089). Strikingly, expression of myr-mApple-PAR3(710-1089) completely rescued the survival of cells depleted of endogenous Par3 (Fig. 8d,e), reversed the drop

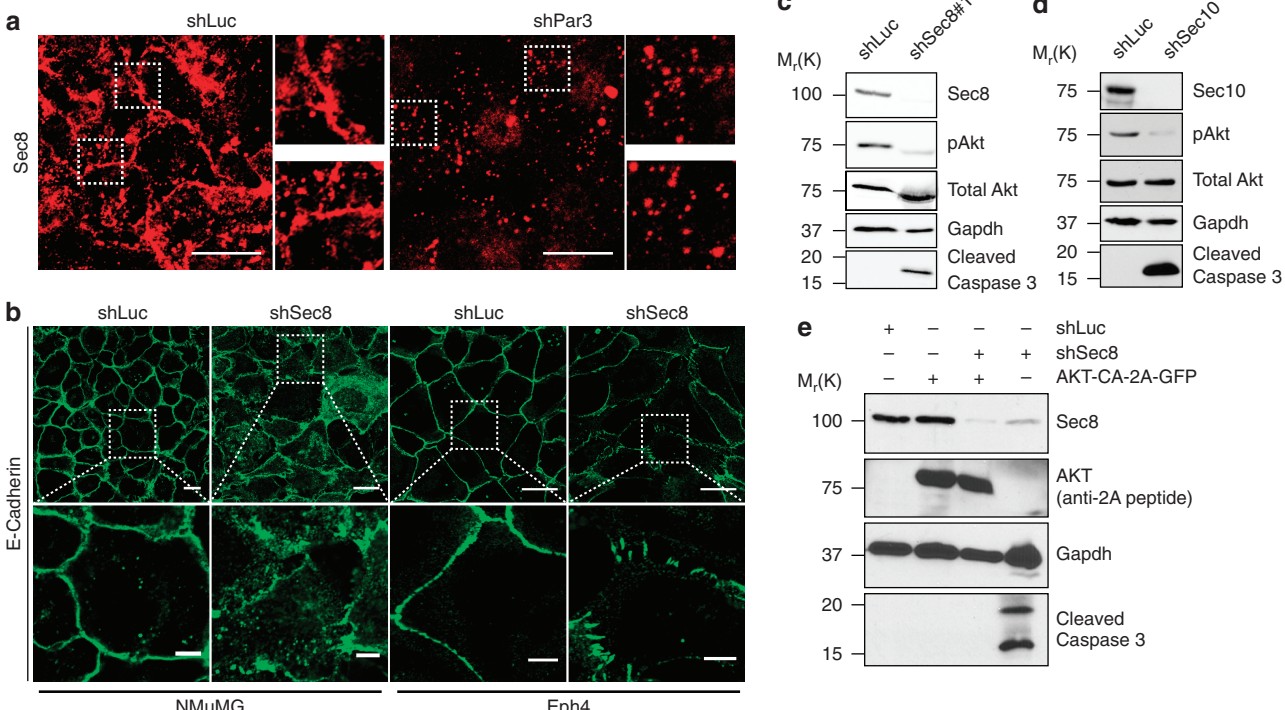

**Figure 6 | Loss of exocyst proteins in NMuMG and Eph4 phenocopies loss of Par3.** (**a**) Localization of endogenous Sec8 in NMuMG cells treated with shLuc or shPar3. Scale bars, 20 μm. (**b**) E-cadherin localization in NMuMG or Eph4 cells treated with shLuc or shSec8. Boxed ROIs are magnified. Scale bars, 20 μm (top), 5 μm (bottom). pAkt and cleaved Caspase-3 levels in NMuMG cells treated with (**c**) shSec8, or (**d**) shSec10. (**e**) pAkt and cleaved Caspase-3 levels in NMuMG cells treated with shSec8 with or without AKT-CA. Experiments were repeated at least three times and representative images are shown.

in pAkt levels (Fig. 8b,c), and partially rescued E-cadherin localization at the cell cortex, although the cells did not appear to be correctly polarized (Fig. 8f). Interestingly, expression of myr-PAR3(710-1089) promoted E-cadherin localization to intercellular junctions even in control NMuMG cells (Fig. 8f). We also observed that Sec6 localized at the junctions in cells expressing shLuc, was disrupted by Par3 depletion, but was restored almost to normal junctional levels upon introduction of the myr-PAR3 exocyst-binding fragment (Fig. 8g–i). We infer that the exocyst-binding domain of Par3 is sufficient, if membrane-associated, to rescue cell survival and the lateral enrichment of both the exocyst and membrane proteins such as E-cadherin.

**The LRD is necessary to rescue cell survival.** Finally, we determined that the deletion of either of two small regions (amino acids 990-1018 or 1014-1043) within the LRD independently were sufficient to disrupt the Par3-exocyst interaction (compare lanes 2–4; Fig. 9a). To determine if these deletion mutants can localize to the PM correctly, we expressed YFP-tagged wild-type PAR3, PAR3(Δ990-1018 or PAR3(Δ1014-1043) in NMuMG cells. All proteins were expressed at equivalent levels and able to accumulate at cell–cell junctions (Fig. 9b,c), but neither PAR3Δ990-1018 nor PAR3Δ1014-1043 was able to rescue cell survival or pAkt levels as robustly as wild-type PAR3 (Fig. 9d–g). Consistent with these results, expression of PAR3Δ990-1018 or PAR3Δ1014-1043 was unable to fully rescue E-cadherin localization at the cell–cell junctions (Fig. 9i,j). Given the defects in adherens junctions we also asked if the Par3-exocyst interaction is important for TJ formation in NMuMG and Eph4 cells. Loss of Par3 severely disrupted the TJs in both cell types, and this could not be rescued by replacing the endogenous Par3 with PAR3b(Δ990-1018) or PAR3(Δ1014-1043) (Fig. 9h).

## Discussion

The exocyst is a highly conserved octomeric complex required for vesicle delivery to regions of the PM actively involved in exocytosis, such as the bud tip in *Saccharomyces cerevisiae*, and the basolateral membranes of epithelial cells[17,19,43]. It tethers vesicles to the PM before SNARE-driven membrane fusions[22,49–52]. Two subunits of the yeast exocyst, Exo70 and Sec3, possess polybasic motifs that can bind $PIP_2$ (refs 25–27), and additional interactions with small GTPases[53–57] have been implicated in the attachment of exocyst to the membrane. In addition, Sec6 may anchor the exocyst to the PM through an unidentified receptor[58,59]. In mammalian epithelia, exocyst delivery has been localized primarily to a region of the lateral membrane adjacent to the TJ[19,23,44]. The mechanism through which this spatial restriction is achieved has not been elucidated but likely requires a TJ-associated receptor, or a chaperone that delivers the exocyst to this region. However, no such receptor/chaperone has been identified to date.

Our work suggests that the polarity protein Par3 can function as an exocyst receptor, and—surprisingly—this function is required for the survival of mammary epithelial cells. Par3 is in the correct location—proximal to the TJs—to act as a receptor, and several studies have reported interactions between polarity proteins and the exocyst, including Par3, Par6 and aPKC. However, functional roles for these interactions have not been elucidated. Moreover, since the three polarity proteins form a complex with one another it has been unclear as to which of them, if any, associate with exocyst directly and which associate indirectly. In a kidney cell line, the scaffold protein Kibra (WWC1) was reported to bridge aPKC to the exocyst, a link required for delivery of the kinase to the leading edge of migrating cells[46]. Others have reported direct binding of Par6 to Exo84 (*Exoc8*) *in vitro*[45]. Another polarity protein, Pals1 (protein

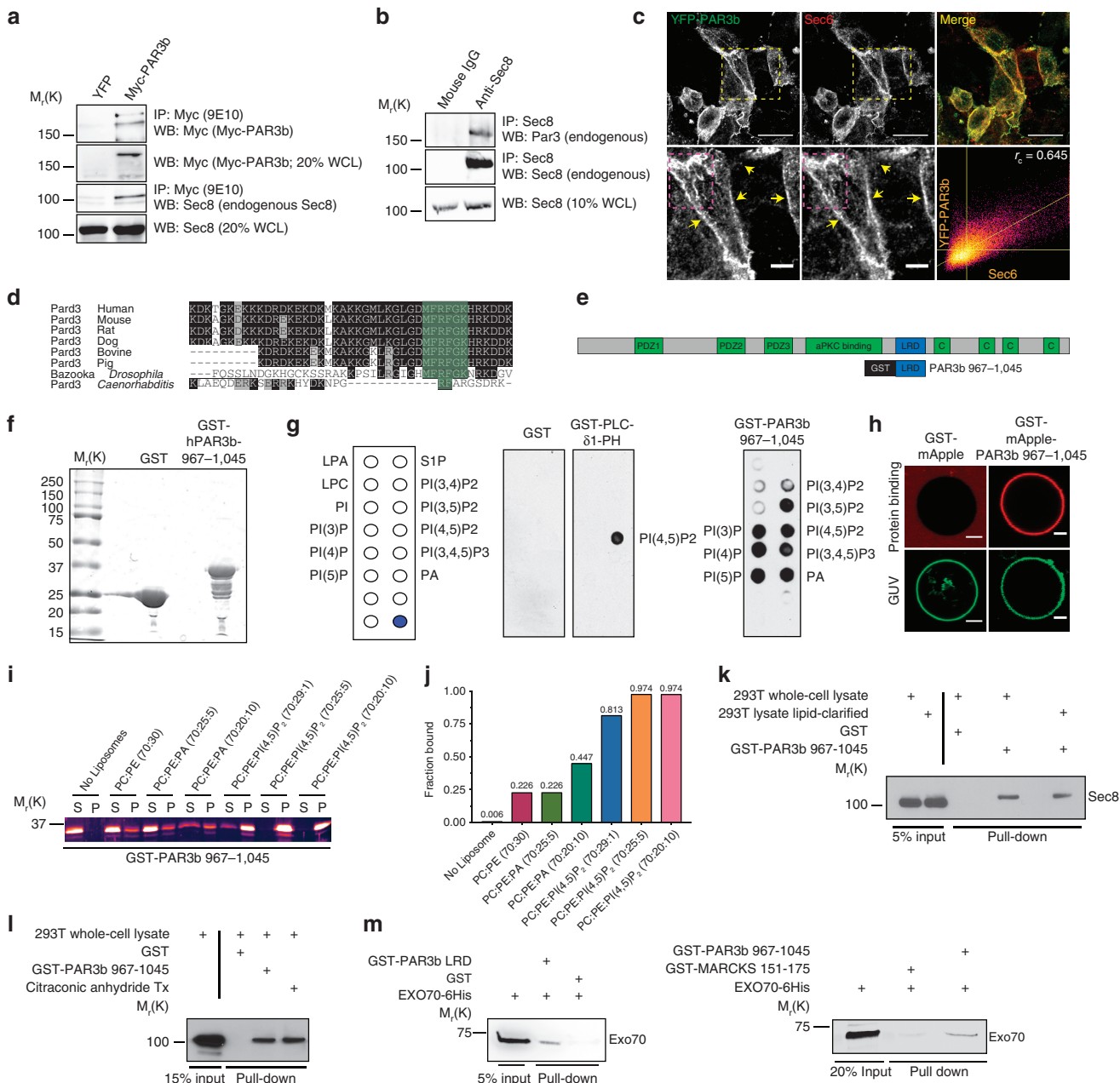

**Figure 7 | The exocyst and phosphatidylinositides bind to the lysine-rich region of Par3. (a)** Co-immunoprecipitation of Myc-PAR3 and Sec8. HEK293T cells were transfected with either pKMyc-PAR3b, or PKYFP as a negative control, and immunoprecipitated with anti-Myc (9E10), antibodies. Lysates were immunoblotted with anti-Myc-HRP and anti-Sec8 antibodies. **(b)** Co-immunoprecipitation of endogenous Par3 and Sec8 from NMuMG cells. Sec8 was immunoprecipitated using anti-Sec8 (14/SEC8) antibody and immunoblotted using anti-Par3 and anti-Sec8 antibodies. Mouse IgG was used as control. **(c)** Localization of YFP-PAR3 and Sec6. NMuMG cells were transduced with YFP-PAR3 (green), grown in a monolayer, fixed and stained for Sec6 (red). Boxed ROIs (yellow) magnified. Pearson correlation coefficient was measured from the boxed ROI (magenta). $r_c$ = Pearson coefficient of co-localization. Scale bars, 20 μm (top), 5 μm bottom. **(d)** Protein sequence alignment of the LRD of Par3 in different species. The green highlighted amino acids are conserved between all species compared. **(e)** Schematic representation showing the domain structures spanning Par3. c = coil-coiled domain. **(f)** Coomassie staining of purified GST and GST-human PAR3(967–1045) from *E. coli*. **(g)** Purified GST, GST-PLC1-PH or GST-PAR3(967–1045) incubated with membrane lipid-strips and immunoblotted with anti-GST2 antibody. **(h)** Confocal images of purified GST-mApple or GST-mApple-PAR3b 967–1045 binding to GUVs. The GUVs consist of DOPC (69 mol%), DOPE (13.5 mol%), PIP2 (1.5 mol%) PA-NBD (0.5 mol%) and cholesterol (15.5 mol%); scale bars, 5 μm. **(i)** Coomassie stained gel (pseudocolored for enhanced contrast) of purified GST-PAR3(967–1045) binding to liposomes consisting of different lipid mixtures as indicated. S = supernatant, P = pellet. **(j)** Levels of protein binding as fraction bound, [P/(S + P)], shown in **i**. **(k)** GST-PAR3(967–1045) binding to SEC8 in native lipid or lipid-free environment. Lipids were removed from purified GST-PAR3(967–1045) and HEK293T lysates using lipid removal adsorbent before mixing and pull-down assay. **(l)** GST-PAR3(967–1045) binding to SEC8 with and without citraconic anhydride treatment. Also, see Supplementary Fig. 7h,i. **(m)** GST capture experiment *in vitro*. Purified GST-PAR3(967–1045), GST or GST-MARCKS 151-175 were mixed with purified EXO70-6His and the associated complex pulled down using Glutathione-Sepharose resins followed by immunoblotting with an anti-EXO70 antibody. Representative images are shown from at least three successful replicates.

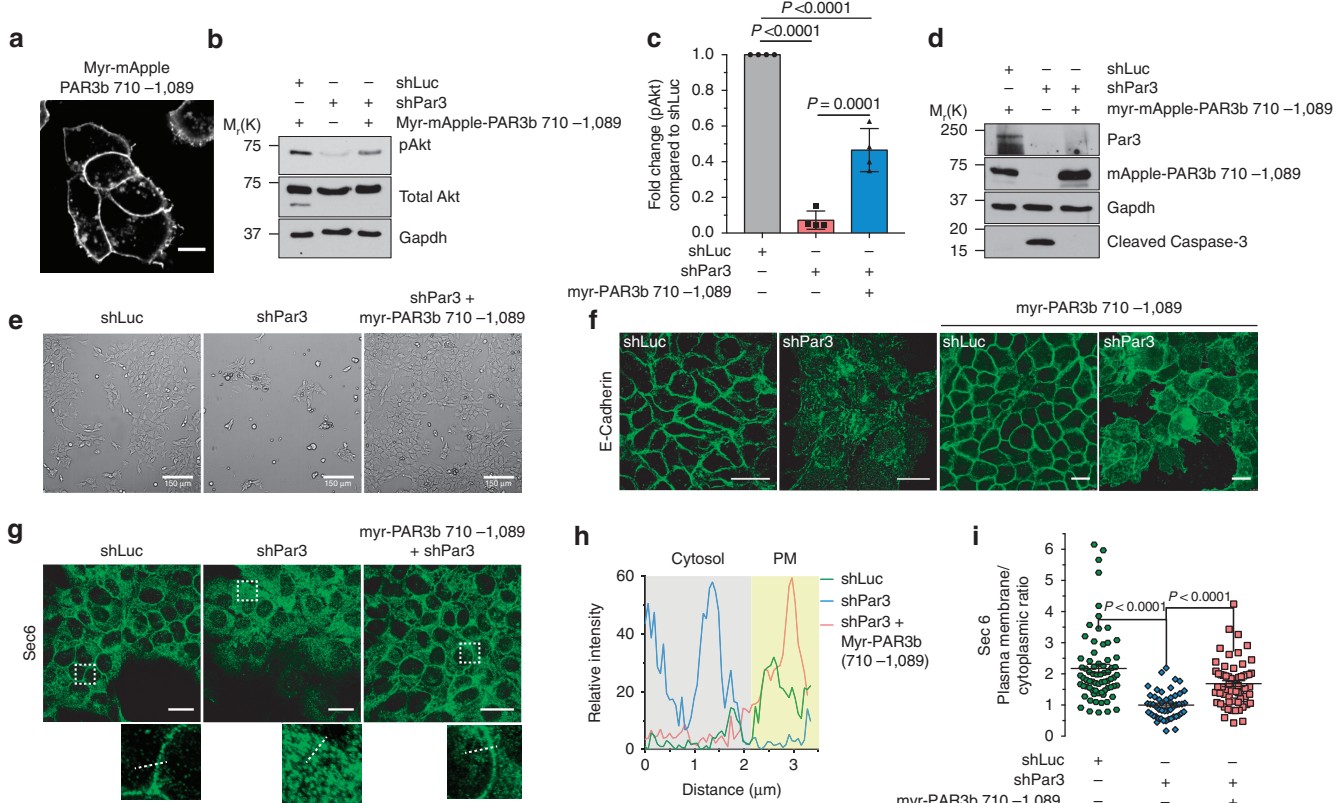

**Figure 8 | Par3-exocyst binding is sufficient for cell survival and E-cadherin localization.** (**a**) Confocal image showing localization of myr-mApple-PAR3(710-1089); scale bar, 10 μm. (**b**) pAkt levels in NMuMG cells expressing shLuc, shPar3 or shPar3 + myr-mApple-PAR3(710-1089). Gapdh and Total Akt used as protein-loading controls. (**c**) Quantification of experiments shown in **b** from three independent experiments, mean ± s.d. *P* values were calculated using one-way ANOVA followed by Dunnett's multiple comparison tests. (**d**) Western blot analysis of cleaved caspase-3 in NMuMG cells expressing shLuc, shPar3 or shPar3 with myr-mApple-PAR3(710-1089). (**e**) DIC images of NMuMG cells expressing shLuc, shPar3 or shPar3 with myr-PAR3b (710-1089); scale bar, 150 μm. (**f**) Confocal images showing E-cadherin localization in NMuMG cells expressing shLuc, shPar3 alone or with myr-mApple-PAR3(710-1089); scale bars, 20 μm. (**g**) Confocal images showing Sec6 localization in NMuMG cells expressing shLuc, shPar3 or shPar3 with myr-mApple-PAR3(710-1089). White boxes represent cropped images shown below; scale bars, 20 μm. (**h**) Intensity plot corresponding to the white lines in the cropped images in **g**. Cytoplasmic portion shaded in grey and approximate PM portion shaded in yellow. (**i**) Quantification of the PM to cytoplasmic ratio of Sec6 localization from three independent experiments; Graph shows mean ± s.e.m., *P* values assessed using one-way ANOVA followed by Kruskal–Wallis multiple comparison tests. ANOVA, analysis of variance.

associated with Lin Seven 1) was described to be important for the polarized localization of Sec8 and syntaxin4, and for the distribution of E-cadherin and myelin proteins at the PM of neurons[60]. In mammary epithelial cells, we find that Par3 binds to the exocyst independently of aPKC. A small fragment of Par3 that cannot interact with either aPKC or Par6 associates robustly with the exocyst through direct association with Exo70. Whether other components of the exocyst components also directly associate with Par3 remains to be elucidated. Most importantly, the expression of Par3 is essential for exocyst function. Silencing of Par3 causes a profound defect in delivery of lateral membrane proteins. The exocyst, normally enriched in the region of the TJ, becomes distributed into internal vesicles. The effects of silencing Par3 expression are phenocopied by depletion of exocyst subunits. Expression of a myristoylated fragment of Par3 that binds to exocyst is sufficient to rescue lateral membrane delivery, whilst the expression of full-length Par3 lacking this domain cannot rescue delivery.

The PDZ1 domain of Par3 has been suggested to recruit the protein to TJs through association with Junctional Adhesion Molecule-A (JAM-A)[61], and interactions with Par6 and aPKC also help recruitment to the junctions[62,63]. Therefore, these interactions provide spatial specificity, enabling the LRD of Par3 to bind phosphoinositides present in adjacent membrane.

Our previous work showed that the defect in TJ formation in MDCK cells as a result of Par3-loss could be rescued by expressing the C-terminal portion of Par3c, a variant which is unable to bind aPKC and missing the first two PDZ domains[8]. Now we show that the exocyst-binding capability of this region is specifically attributable to this function, because expression of PAR3b(Δ990-1018) or PAR3(Δ1014-1043) in Par3-depleted NMuMG or Eph4 cells was not able to rescue TJs. We propose that the interaction of PIP$_2$ with Par3 assures close proximity by the exocyst to the PM. Exo70 and Sec3 can also bind phosphoinositides, providing an avidity effect, which will be further enhanced by the oligomerization of Par3 through the N-terminal CR1 domain (Fig. 10). The isolated, myristoylated, exocyst-binding domain of Par3 lacks the spatial specificity of the full-length Par3, but can enable exocyst docking sufficiently to rescue cell survival. These results are interesting in the light of previous observations that the lipid kinase PIPKIγ can bind both to Exo70 and E-cadherin, generating PIP$_2$ pools at nascent E-cadherin contacts[64]. In this model, Par3 would provide the landmark that triggers an initial delivery of E-cadherin, and this is reinforced by a positive feedback loop, in which local PIP$_2$ production further promotes Par3 and exocyst recruitment, enhancing delivery of E-cadherin to establish stable adherens junctions.

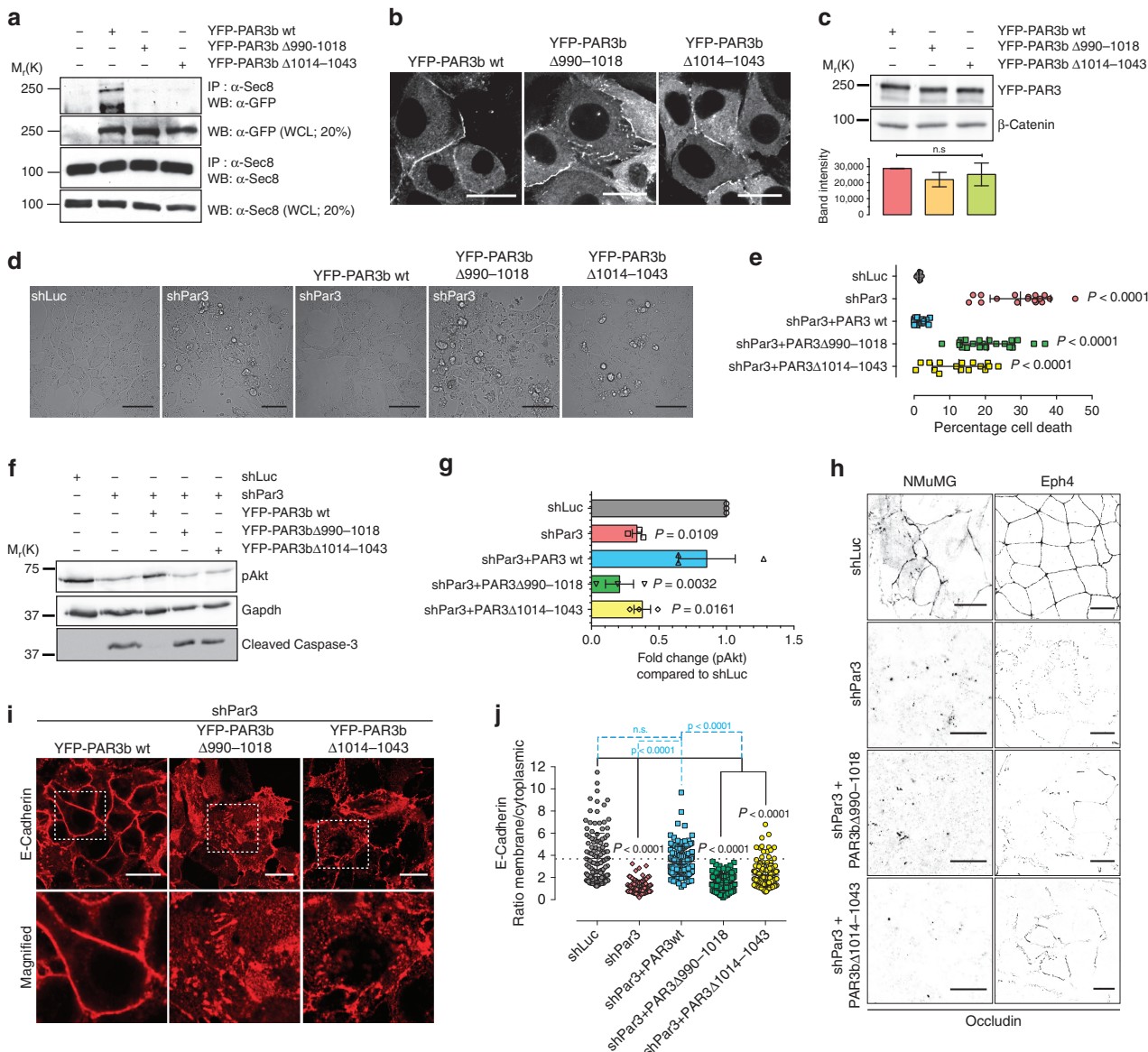

**Figure 9 | Par3-exocyst interaction is necessary for cell survival.** (**a**) Interaction of SEC8 with wild-type YFP-PAR3, YFP-PAR3Δ(990-1018) or YFP-PAR3Δ(1014-1043). YFP-PAR3 constructs were transiently expressed in HEK293T cells and immunoprecipitated using an anti-Sec8 antibody, followed by immunoblotting with anti-GFP or anti-Sec8. (**b**) Confocal images showing localization of wild-type YFP-PAR3, YFP-PAR3Δ990-1018 or YFP-PAR3Δ1014-1043 in NMuMG cells; scale bars, 20 μm. (**c**) Western blots showing expression levels of YFP-Par3, YFP-PAR3Δ990-1018 or YFP-PAR3Δ1014-1043 expressed stably in NMuMG cells. β-catenin was used as loading control. Quantifications from three independent experiments and shown as average band intensity. Error bars, mean ± s.e.m. (**d**) Representative DIC images of NMuMG cells expressing shLuc, shPar3 or shPar3 with wild-type YFP-PAR3b, YFP-PAR3bΔ(990-1018) or YFP-PAR3Δ(1014-1043) from three independent experiments. Scale bars, 50 μm. (**e**) Quantification of the percentage of cells dying; the graph shows mean ± s.d. $n = 3$. (**f**) Immunoblot showing pAkt and cleaved caspase-3 levels in NMuMG cells expressing shLuc, shPar3 or shPar3 with wild-type YFP-PAR3b, YFP-PAR3bΔ(990-10180 or YFP-PAR3Δ(1014-1043). (**g**) Quantification of three experiments shown in **f**. Error bars, mean ± s.e.m. (**h**) Confocal images showing TJs in NMuMG and Eph4 cells. Wild-type cells or cells expressing YFP-PAR3bΔ(990-1018), or YFP-PAR3Δ(1014-1043) were transduced with shLuc or shPar3 lentivirus followed by paraformaldehyde fixation and staining with an anti-Occludin antibody. Scale bars, 20 μm. Image colours were inverted for a clearer depiction of Occludin structures. (**i**) E-cadherin localization in NMuMG cells expressing shPar3 with wild-type YFP-PAR3b, YFP-PAR3bΔ(990-1018) or YFP-PAR3Δ(1014-1043). Scale bars, 20 μm. The white dashed box represents the magnified region in the images. (**j**) Quantification of the PM/cytoplasmic ratio of E-cadherin; graph shows mean ± s.d. Dotted line represents the average for shLuc expressing NMuMG cells. All experiments successfully repeated three times. $P$ values were computed using one-way ANOVA followed by Dunnett's multiple comparison tests.

Interestingly, in the *C. elegans* excretory cell, Par3 concentrates membrane-localized exocyst proteins to a specific polarized domain at the lumenal surface, through an unknown mechanism[65]. Although the Par3 sequence of *C. elegans* is not highly homologous to the mammalian protein in the LRD, these observations suggest that our proposed receptor mechanism is likely conserved throughout the animal kingdom. However,

further experiments are needed to investigate if the *C. elegans* Par3 homologue directly binds to the exocyst.

The consequences of defective exocyst function are profound in mammary epithelial cells. The key effect is a reduction in basolateral $PIP_3$ formation, probably because E-cadherin is not delivered to the PM, resulting in mislocalization of PI-3 K. Reduced $PIP_3$ leads to decreased phosphorylation of Akt, which

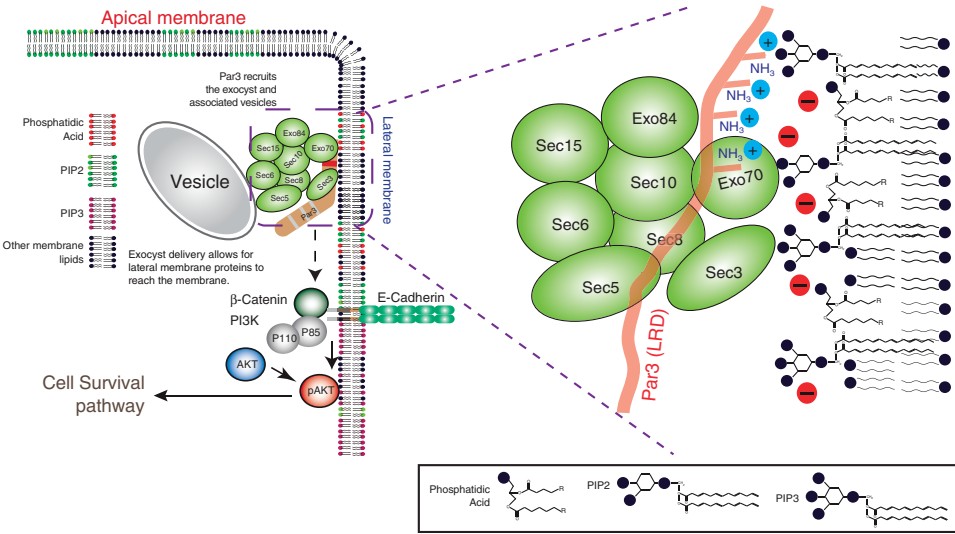

**Figure 10 | Model for the role of Par3 as an exocyst receptor.** Schematic showing the role of Par3 as an exocyst receptor and how it is the required for mammary cell survival. The magnified schematic representation shows that the positively charged lysine residues of Par3 interacts with the negatively charged phospholipids, and possibly recruits Exo70 at the site to dock exocyst onto the PM.

triggers apoptosis. Therefore, the Par3 polarity protein functions as an important survival factor for mammary epithelial cells. These new data are consistent with our previous *in vivo* observations that depletion of Par3 in the mammary progenitor cells resulted in severely reduced mammary gland regeneration, accompanied by increased apoptosis and proliferation[66]. However, the level of apoptosis observed in the regenerated glands was lower than what we observe *in vitro*—possibly because of rapid engulfment of apoptotic cells, or the impact of neighbouring myoepithelial cells or stromal cells in the tissue, or systemic factors that suppress apoptosis. It is important to recognize that although apoptosis wins out in untransformed cells, we and others have shown that loss of Par3 in oncogene-transformed cells, which can evade apoptosis, reveals a hyperproliferative and invasive response that enables rapid tumour growth and metastasis[15,30,67]. We also note that loss of Par3 promotes apoptosis in the mouse epidermis[15]. Nonetheless, *pard3* knockout mice can survive until about E10.5. It is possible that a truncated version of the Par3 protein containing the exocyst-binding domain is still produced in these mice, in which exon3 was deleted, or that another protein, such as the related Par3-like protein (*pard3b*), can also function as an exocyst receptor and compensate for loss of Par3. An important goal for the future will be to understand the regulation of the Par3/exocyst interaction, and to identify other possible exocyst receptors.

## Methods

**Plasmid constructs and other reagents.** The pLVTHM-YFP-hPAR3 construct used here was described previously[7]. To make YFP-PAR3(R596D,K598D), residues R596 and K598 of human PAR3 were converted to aspartates by site-directed mutagenesis PCR. Likewise, all PAR3 mutants used herein were also generated by site-directed mutagenesis using Phusion High-Fidelity DNA polymerase (New England Biolabs). PLVTHM-mApple was generated from pLV-YFP, where YFP was replaced with mApple with a C-terminal multiple cloning site containing: BamHI, NotI, EcoRI, SpeI and NdeI restriction endonuclease sites. A pLVTHM-myr-mApple vector was created by adding the myristoylation sequence MGSSKSKPKDPSQRRRRIRGYL at the N-terminus of the mApple coding sequence. A constitutively active AKT1 lentiviral construct (AKT-CA-2A-GFP) was generated in the pLVTHM vector. The coding sequence from the plasmid HA-AKT1(T308D, S473D), which was a gift from Jim Woodgett (Addgene plasmid 14751)[34] was PCR amplified and inserted in the BamHI/EcoRI cassette in frame with the self-cleaving 2A peptide and with GFP at its C-terminus[68]. To generate the PIP₃ sensor PH-AKT, the PH domain of human AKT1 was inserted between the BamHI/EcoRI sites of the pWPI vector to generate a lentiviral YFP-PH-AKT

construct. To make pLV-YFP-RAB8 and RAB11, RAB8 and RAB11 (gifts from Dr Stephane Angers and Dr Terrence Hébert) were PCR amplified and inserted into the BamHI/NotI cassette in pLVTHM-YFP vector. To express hPAR3(967–1045) in bacteria, the fragment was cloned into pGEX-4 T in frame with GST as the N-terminal fusion protein. To generate GST-MARCKS 151-175, MARCKS aa151-175 was synthesized as a geneblock (IDT, Inc) and cloned into pGEX-4 T by Gibson assembly, downstream of the GST tag. All other plasmids used were described previously. The VSVG-ts045-GFP construct was a gift from Jennifer Lippincott-Schwartz[38]. 800 pSG5L HA-PTEN was from William Sellers (Addgene plasmid #10750)[69].

Mouse Par3 shRNA (5′-GTAGGCAAGAGGCTCA-3′) was described previously[7]. The following shRNA clones were purchased from the Sigma MISSION shRNA library: mouse Sec8 shRNA clones TRCN0000307390 and TRCN0000298307, mouse Sec10 shRNA clone TRCN0000093547, mouse Pten shRNA clones TRCN0000322421 and TRCN0000322487 and mouse aPKCλ shRNA clone TRCN0000278129. Knockdown efficiencies with the shRNAs were determined by immunoblot of the endogenous protein levels after treatment with shRNA lentiviral particles compared with a shRNA towards the luciferase gene.

For immunoblotting, we used the following antibodies: rabbit anti-Par3 (refs 7,30), rabbit anti-cleaved Casp3 (Asp175) (1:1,000, Cell Signaling Technology, Clone 5A1E), rabbit anti-Parp (1:1,000, Cell Signaling Technology), mouse anti-β-tubulin (1:1,000), mouse anti-Myc (1:1,000, clone 9E10), rabbit anti-Gapdh (1:1,000, Cell Signaling Technology, clone 14C10), rabbit anti-phospho-Foxo3a (Ser253) (1:1,000, Cell Signaling Technology), rabbit anti-Bim (1:1,000, Cell Signaling Technology, clone C34C5), rabbit anti-phospho-Bad (Ser136) (1:1,000, Cell Signaling Technology, clone D25H8), rabbit anti-Bad (1:1,000, Cell Signaling Technology), rabbit anti-phospho-Akt (Ser473) (1:1,000, Cell Signaling Technology, clone D9E), rabbit anti-Akt (1:1,000, Cell Signaling Technology), mouse anti-β-catenin (1:2,000, BD Biosciences, clone 14/Beta-Catenin), rabbit anti-E-cadherin (1:1,000, Cell Signaling Technology, clone 24E10), mouse anti-GST (1:2,000, clone GST2), rabbit anti-RFP (1:1,000, Rockland), rabbit anti-2A peptide (1:1,000, EMD Millipore), rabbit anti-GFP (1:1,000), chicken anti-GFP (1:1,000, Abcam), rabbit anti-Pten (1:1,000, Cell Signaling Technology, clone 138G6), rabbit anti-aPKCζ (1:1,000, Santa Cruz Biotechnology). For immunofluorescence we used the following antibodies: rabbit anti-cleaved Casp3 (Asp175) (1:100, Cell Signaling Technology, Clone 5A1E), chicken anti-GFP (1:250, Abcam), rabbit anti-E-cadherin (1:200, Cell Signaling Technology, clone 24E10), mouse anti-E-cadherin (1:200, BD Bioscience, clone 36/E-cadherin), mouse anti-Sec8 (1:100, BD Biosciences, 14/SEC8), mouse anti-Sec6 (1:100, Novus Biologicals, clone 9H5), rabbit anti-Sec10 (H-300) (1:1,000, Santa Cruz Biotechnology), rabbit anti-phospho-Akt (Ser473) (1:200, Cell Signaling Technology, clone D9E),, rabbit anti-β-catenin (1:500; Sigma). mouse anti-Zo1 (1:500, Life Technologies, clone Zo1-1A12), and mouse anti-Occludin (1:500, ThermoFisher Scientific). Mouse-anti-NKA antibody (1:100, clone 6H) was a gift from Michael Caplan. CellEvent Caspase-3/7 Green ReadyProbes Reagent, Alexa-647 Phalloidin, NucBlue Live ReadyProbes, Hoechst 33342 and 7-Aminoactinomycin D (7-AAD) were all from Life Technologies. AnnexinV Apoptosis Detection Kit was from eBioscience. BFA, Wortmannin, and LY294002 were from Cell Signaling Technology. Ac-DEVD-CHO Casp3 inhibitor was from BD Pharmingen. MK-2206 was purchased from Selleck Chemicals. 1,2-dioleoyl-*sn*-glycero-3-phosphochloine (DOPC), 1,2-dioleoyl-*sn*-glycero-3-phosphoethanolamine (DOPE), 1-Stearoyl-2-Oleoyl-sn-

Glycero-3-phosphate, and 1,2-dioleoyl-sn-glycero-3[phosphoinositol-4,5-bisphosphate] were purchased from Avanti Polar Lipids.

**Stable cell lines.** NMuMG or Eph4 cells stably expressing YFP-PAR3 or YFP-PAR3 mutants were established by lentiviral transductions followed by fluorescence-activated cell sorting using the YFP fluorochrome as the marker. Cells were sorted using BD FACSAria III cell sorter. Similarly, YFP-RAB8 and YFP-RAB11 constructs were expressed using lentiviral transductions; however, cells were not sorted before performing experiments.

**Cell culture and lentiviral transductions and transfections.** NMuMG and HEK293T cells were obtained from ATCC. Eph4 cells were obtained from Dr. Jürgen Knoblich (Institute of Molecular Biotechnology, Vienna, Austria). NMuMG, Eph4, and HEK293T cells were cultured in Dulbecco's Modified Eagle Medium (Life Technologies) supplemented with 10% fetal bovine serum (Atlanta Biologicals), and $1 \times$ Penicillin/Streptomycin and Glutamine (Life Technologies) and maintained in culture as suggested by ATCC. Lentivirus was produced by transfecting HEK293T cells with the lentiviral packaging vectors psPAX2, pMD2.G using calcium phosphate precipitation. All lentiviral transductions for protein expression were performed at an MOI of 5, and all shRNA infections at an MOI of 20 (based on titres measured using 293T cells). NMuMG cells were transfected using Xfect transfection reagent (Clontech). Transfections in HEK293T cells for co-immunoprecipitation experiments were performed using calcium phosphate precipitation.

**Flow cytometry analysis of apoptotic cells.** Annexin V binding to cells was assessed to quantify cellular apoptosis by flow cytometry. Cells were harvested by trypsinization to obtain single cell suspensions. Cells were washed with $1 \times$ PBS followed by $1 \times$ binding buffer provided with the Annexin V apoptosis detection kit (eBioscience). Cells were resuspended in $1 \times$ binding buffer at a density of $2 \times 10^6$ cells per ml; 5 μl of Annexin V-APC was added to 100 μl of the cell suspension and incubated at room temperature for 15 min with rocking, protected from light. Following antibody labelling, cells were washed twice with 1 ml of $1 \times$ binding buffer and resuspended in 200 μl of $1 \times$ binding buffer together with 7-AAD. Samples were analysed by flow cytometry using the Guava easyCyte 8HT flow cytometer (EMD Millipore) operated by GuavaSoft version 2.6. Data acquired were analysed using FlowJo version 9.6.

**Isolation of mouse primary luminal mammary epithelial cells.** All mice were housed and handled according to protocols approved by the Institutional Animal Care and Use Committee of Vanderbilt University to comply with ethical regulations. The third and fourth mammary gland pairs were removed from 8-week-old C3H female mice, minced thoroughly with scissors and digested in freshly prepared Digestion media containing DMEM/F12, 2 mg ml$^{-1}$ Collagenase I (Roche), 5 ug ml$^{-1}$ insulin (Sigma), 600 U ml$^{-1}$ Nystatin (Sigma), 100 U ml$^{-1}$ penicillin/streptomycin for 1 h at 37 °C with mixing. The epithelial organoids were collected by centrifugation at 450 g for 5 min. The cells pelleted were resuspended in 5 ml of DMEM/F12 and centrifuged at 450 g for 15 s. The supernatant was removed and this step was repeated five times. Doing so enriches for epithelial organoids. After the final wash, cells were resuspended in 0.05% Trypsin/EDTA and incubated at 37 °C with gentle shaking for 12–15 min followed by addition of 2 U ml$^{-1}$ of DNase I for 1 min. The trypsin and DNase I activity was blocked by adding 0.5 ml of calf serum. Cells were pipetted up and down a several times to dissociate any residual clumps and filtered with a 40 μm cell strainer to collect single cells. After the isolation of purified epithelial organoids and forming single cell suspensions, cells were resuspended in ice-cold PBS containing 10 mM HEPES, 2 mM EDTA and 2% fetal bovine serum. Cells were then stained for 10 min with mouse anti-CD326(EpCAM)-APC (1:200; eBioscience), rat anti-CD49f-PerCP/Cy5.5 (1:200; Biolegend). Mouse anti-CD45-PE, rat anti-mouse CD31-PE and rat anti-mouse Ter119-PE were also added to remove any non-epithelial lineage cells. After washing once, cells were resuspended in the above buffer and supplemented with DAPI (1 ug ml$^{-1}$) and sorted by FACS (BD FACSAria Ill) for EpCAM$^{high}$/CD49f$^{med}$ cells as shown in Supplementary Fig. 3. Sorted cells were plated on laminin coated (1 ug ml$^{-1}$) plates and grown in freshly prepared MEC medium containing DMEM/F12, $1 \times$ Insulin-transferrin-selenium (Sigma), 5 ng ml$^{-1}$ EGF, 100 U ml$^{-1}$ Penicillin/Streptomycin, 20 U ml$^{-1}$ Nystatin, 5 ml fetal bovine serum, and $1 \times$ Glutamax (Gibco) for subsequent experiments.

**Measurement of PIP$_3$ levels.** PM levels of PIP$_3$ were assessed using PH-AKT-GFP as a sensor. A stable polyclonal cell line was generated by transducing NMuMG cells with lentivirus expressing PH-AKT-GFP. Cells were then transduced with shRNAs targeting Luciferase (control) or Par3 in the presence of Ac-DEVD-CHO (50 μM). Three days after transduction cells were fixed with 4% paraformaldehyde (EMD Millipore) and imaged using a $63 \times$/1.40 Plan-APOC-HROMAT oil immersion lens on Zeiss LSM710 confocal microscope. Fluorescence intensities of PH-AKT-GFP at the PM and in the cytoplasm were measured from the regions of interests (ROI) of cells using ImageJ software (ver 1.46r for Mac) and reported as a frequency distribution histogram of membrane to cytoplasmic ratio.

As an alternative, NMuMG cells transduced with shLuc or shPAR3 were grown on 150 cm$^2$ culture plates in the presence of Casp3 inhibitor Ac-DEVD-CHO (50 μM). To extract PIP$_3$, the medium was removed from cells by gentle aspiration followed by immediately adding 10 ml of ice-cold 0.5 M trichloroacetic acid (TCA). Cells were collected by scraping followed by centrifugation at 300 g for 5 min. Pellets were washed with 3 ml of 5% TCA/1 mM EDTA. Neutral lipids were extracted by adding 3 ml of methanol:CHCl$_3$ (2:1), vortexing $3 \times$ over 10 min at room temp followed by centrifugation at 300 g for 5 min. Next, 2.25 ml of Methanol: CHCl$_3$: 12 M HCl (80:40:1) was added to the pellet and vortexed $4 \times$ over 15 min at room temp. Supernatant was collected by centrifugation at 300 g, followed by treatments with 0.75 ml of CHCl$_3$ and 1.35 ml of 0.1 M HCl, vortexed and centrifuged at 300 g for 5 min to separate organic and aqueous phases. Lipid from the lower organic phase was collected, vacuum dried and used for subsequent ELISA assay to assess total cellular PIP$_3$ levels using PIP$_3$ Mass ELISA kit (Echelon) according to manufacturer's protocol.

***In vitro* protein expression.** To express recombinant PAR3 fragments *in vitro*, the desired coding sequences were cloned into the pGEX-4 T vector downstream of GST. Proteins were expressed in BL21 (DE3) competent *E. coli* by induction with isopropyl 1-thio-β-D-galactopyranoside (500 μM; Sigma) in a shaker at room temp (240 r.p.m.) overnight. The bacterial pellet was lysed by sonication in PBS supplemented with 1 mM phenylmethylsulfonyl fluoride. GST-tagged proteins were purified using Glutathione-Sepharose 4B (GE Healthcare) resins at room temp for 1 h with end over end mixing. The bound proteins were washed $3 \times$ with PBS and kept on beads for further protein binding experiments or eluted with 10 mM reduced glutathione/50 mM Tris-HCL pH 8.0 buffer for lipid-binding experiments. Eluted proteins were concentrated and buffer exchanged into PBS using an Amicon Ultra-0.5 Centrifugal Filter unit with membrane nominal molecular weight limit (NMWL) of 10 kDa (EMD Millipore).

**Lipid-binding assay.** Lipid–protein interaction experiments using PIP Strips (Echelon Biosciences) according to the manufacturer's protocol. Purified GST protein and PtdIns(4,5)P$_2$ Grip (PLC-δ1-PH) (Echelon Biosciences) were used as negative and positive controls respectively. All protein–lipid-binding experiments were carried out with 0.5 μg ml$^{-1}$ of proteins in PBS-T with 3% BSA. Protein binding to lipids was assessed by immunoblotting using mouse-anti-GST2 antibody.

For liposome preparations, lipids as indicated were dried on the bottom of glass tubes and hydrated in 20 mM Hepes pH 7.4 to yield 1 mg ml$^{-1}$ samples. Hydrated lipids were vortexed followed by 10 freeze-thaw cycles and sonication in a water sonicator bath for 1 min to form small unilamellar vesicles. 100 μl of 1 mg ml$^{-1}$ liposome sample was mixed with recombinant GST-Par3(967–1045) protein (∼30 nM) in a polycarbonate ultracentrifuge tube (Beckman Coulter Inc.) and spun at 150,000 g in a TLA-100 rotor using a TL-100 micro-ultracentrifuge (Beckman Coulter Inc.). The pelleted and supernatant fractions were then run on SDS–polyacrylamide gel electrophoresis and assessed by either Coomassie staining or western blot analysis.

GUV were electroformed on Indium-Tin-Oxide-coated (ITO) glass coverslips (Sigma Aldrich). Lipid mixture (20ul; 10 mg ml$^{-1}$) containing DOPC (69 mol%), DOPE (13.5 mol%), PIP2 (1.5 mol%) PA-NBD (0.5 mol%) and cholesterol (15.5 mol%) was dried on each side of the ITO slide under a stream of nitrogen followed by vacuuming in the dark to preserve nitrobenzoxadiazole (NBD) fluorescence. Upon drying the slides were sandwiched with a silicon spacer in between (Electron Microscopy Sciences) with 550 ul of GUV formation buffer (20 mM HEPES pH 7.4, 500 mM Sucrose). The conductive surface of the ITO slides was attached to conductive copper tapes and connected to a function generator (PI-9587C; PASCO Scientific) to create a 2.5–5 V of sinusoidal waveform at 10 Hz, which was monitored using an oscilloscope (2120B; BK Precision). Upon forming the vesicles over 10–12 h, vesicles were mixed with either purified GST-mApple or GST-mApple-Par3(967–1045) (∼150 nM) for 20–30 min and protein–lipid binding observed with a CFI Plan Fluor $40 \times$/1.30 oil immersion lens using a Nikon A1R+ confocal system (Nikon Instruments Inc).

**Western blots and protein interaction experiments.** To detect cleaved Casp3 or other related signalling events cells both attached and floating were collected and lysed in $2 \times$ Laemmli buffer containing β-mercaptoethanol (Sigma-Aldrich), resolved by SDS–polyacrylamide gel electrophoresis, and transferred onto nitro-cellulose (Perkin Elmer) or polyvinylidene difluoride (EMD Millipore) membranes for western blot analysis. All of the original uncropped western blots for the data in this paper are provided in Supplementary Fig. 8.

For immunoprecipitation experiments, cells were lysed in buffer containing 0.1% Triton-X100, 20 mM HEPES (pH 7.4), 50 mM NaCl, 2 mM EDTA supplemented with protease inhibitors cocktail (Roche) and phosphatase inhibitors (Roche). Insoluble materials from the lysates were removed by centrifugation at 16,100 g for 10 min and co-immunoprecipitations of protein complexes were carried out by incubating the lysates with mouse anti-Sec8, rabbit anti-Par3 or mouse anti-MYC antibodies for 2–3 h together with GammaBind Plus Sepharose (GE Healthcare) beads. All protein quantifications were done using Pierce BCA protein assay kit (Thermo Scientific).

*In vitro*, binding experiments were done with recombinant GST-PAR3(967-1089) fragments. HEK293T cells were lysed in buffer containing 1% CHAPS, 20 mM HEPES, 50 mM NaCl, 2 mM EDTA, supplemented with protease or phosphatase inhibitors. All binding experiments were done at 4 °C with end-over mixing for 2 h followed by washing in 1% CHAPS lysis buffer. To block the amine group of lysine within Par3b 967-1089 we used citraconic modification of GST-PAR3(967-1089) in an overnight reaction of protein immobilized on Glutathione beads with the citraconic anhydride in 0.1 M sodium carbonate buffer, pH 8 (Thermo Scientific). Protein binding to glutathione beads was not affected by citraconic anhydride treatment. Amine blocking was verified using a fluorimetric assay with O-Phthaldialdehyde (Sigma-Aldrich)[70]. Lipids from cells and bacterial lysates were removed using CleanasciteTM Lipid Removal Reagent and Clarification (Biotech Support Group LLC).

**Immunofluorescence and image acquisition.** Cells grown on LabTek II chamber slides (Thermo Scientific), or on No. 1.5 coverglass (Thermo Fisher Scientific) were fixed with 4% paraformaldehyde-PBS, pH 7.4 or in Methanol at − 20 °C for 15 min. When fixed in paraformaldehyde, cells were permeabilized with 0.5% Triton X-100. Fixed cells were treated with 1 × Western Blocking Reagent (Roche Life Sciences) before incubation at 4 °C with the indicated antibodies diluted in Western Blocking Reagent. Secondary labelling was performed using AlexaFluor (Life Technologies) labelled secondary antibodies as indicated. Coverslips were mounted on slides using Fluoromount G (Electron Microscopy Sciences) or Pro-Long Gold antifade mountant (ThermoFisher Scientific).

Laser scanning confocal images were acquired using a Plan-Apochromat 63 × /1.4 NA oil immersion objective on a Zeiss LSM710 META inverted confocal microscope (Carl Zeiss Microscopy), HCX PL APO 63 × /1.4 oil immersion objective on a Leica SP5 confocal microscope (Leica Microsystems) or CFI Apo Lamda S LWD 40X/1.15 water immersion lens, CFI Apo VC 100 × /1.40, or CFI Apo TIRF 60 × /1.49 oil immersion objective using a Nikon A1R+ confocal system (Nikon Instruments Inc). Epifluorescence images were acquired using a Nikon Plan 40 × /0.65 objective using a Nikon Eclipse Ti microscope equipped with Andor Neo sCMOS camera and operated by NIS-Elements Advanced Research imaging software. DIC and phase contrast images were taken using either a Nikon Eclipse Ti or EVOS FL (Life Technologies) inverted microscopes respectively.

VSVG-GFP trafficking experiments were performed in live cells at 40 °C or 32 °C as described previously[38]. Confocal live cell imaging was performed on a Nikon A1R+ microscope using a CFI Apo Lambda S 40 × /1.15 water immersion objective mounted on a Nikon Eclipse Ti microscope stand. For live imaging cells were treated with Prolong Live Antifade (Life Technologies) in phenol-free FluoroBrite DMEM (ThermoFisher Scientific) culture medium to reduce photobleaching and background fluorescence.

Uncompressed images were minimally processed or cropped using ImageJ software (ver 2.0.0-rc-43 for Mac) (National Institutes of Health), or NIS-Elements Advanced Research (Nikon Instruments Inc). To assess membrane cytoplasmic protein localizations ROIs were drawn on the entire cytoplasmic compartment or on membrane regions and pixel intensities were measured using ImageJ/FIJI. To assess co-localization, ROIs were drawn in cells and Pearson correlation measured using the co-localization threshold function of ImageJ/FIJI. The Pearson correlation coefficient of co-localization is given as $r_c$ values embedded in the regression graphs.

**Statistical analysis.** Data are generally reported as ± s.e.m. or ± s.d. and analysed by Student's *t*-test, or one-way analysis of variance as indicated using Graphpad Prism 6 software. When using analysis of variance, *post hoc* analysis was done using Kruskal–Wallis, Tukey or Dunnett multiple comparison tests. All statistical analysis was considered significant at $P < 0.05$.

**Data availability.** All data that support the findings of this study are available within the article and its Supplementary Information, or from the authors upon reasonable request.

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

## Acknowledgements

This work was supported by a NIH/NIGMS grant (5RO1 GM070902-10) to I.G.M. and a Postdoctoral Fellowship to S.M.A. from the Canadian Institutes of Health Research (CIHR).

## Author contributions

S.M.A. designed, performed and analysed the data. I.G.M. designed experiments. S.M.A. and I.G.M. wrote the paper.

## Additional information

**Competing interests:** The authors declare no competing financial interests.

