## [Peer Review File · Nature Communications]

Reviewer #1 (Remarks to the Author):

The manuscript describes a novel function of PAR3 as an exocyst receptor. The authors used mammary cancer cell lines to find that PAR3 depletion causes apoptosis. Initiated by this novel finding, they provide evidence supporting that the primary defect caused by PAR3 depletion is the mislocalization of lateral proteins. They also show that a sequence in PAR3 and its direct association with SEC8, a component of the exocyst complex, are involved in the delivery of a set of cell surface proteins.

Many of the data presented are clear and interesting, and will reveal a novel molecular mechanism how proteins are delivered to their specific intra-cellular location. However, there are many points that should be clarified before publication.

The title of the paper sounds as it reports two points; PAR3 is an exocyst receptor and the receptor is essential for mammary cell survival. However, this reviewer strongly feels that the latter is not fully supported by experimental evidence and that the title might be misleading. Thus, I suggest that the title should be changed. Specific points follows:

1. Previous studies on a variety of species such as *Drosophila* and mammals suggested that PAR3 is involved in localization of apical proteins. What does the central conclusion of the present manuscript mean in the light of the role of PAR3 on localization of apical proteins.

2. Does NMuMG and Eph4 cells show polarized phenotype? Do they have distinct lateral and apical domains? Do they have tight junctions?

3. Does PAR3 depletion affect localization of E-cad and Na⁺-K⁺-ATPase (NKA) as cancer cell lines?

This question relates to the point that most of the data were obtained from cancer cell lines. This reviewer cannot be confident that the very interesting conclusion reflects the physiological role of PAR3 in mammary epithelial cells. In other words, I am afraid that the critical data might be artifacts of the cancer cell line. Since the biochemical data looks strong, the functional interaction between exocyst and PAR3 seems very conceivable. However, there is a possibility that the target of the PAR3-exocyst might be different depending on the cell types. If this is the case, the finding on the role of PAR as a exocyst receptor is quite important.

4. The only data for primary mammary epithelial cells is the one shown in Fig2e. However, the data is not clear; some bands are deformed suggesting sample overloading, casting doubts for the authors conclusion.

5. There is no description how the M/C ratio of PH-AKT-GFP is quantified from the 2D staining data in Fig2i.

(Cf. the confocal data in Figure 6a is clear but those of Fig2i are not clear. And the authors state similar conclusion. Are there any difference in the method?)

6. The authors use the word "survival or apoptosis" based on the appearance of bands of cleaved caspase-3 in most cases. This is misleading and they should precisely describe their data. (Texts for Fig 2g, Fig2j, and Fig2k; Fig4e)

7. Does the LRD sequence motif conserved for *C. elegans* PAR3 (Fig5d)?

Reviewer #2 (Remarks to the Author):

This study starts with the observation that depleting cells of Par3 leads to apoptosis and then traces the effects back to a role for Par3 in recruiting the exocyst to tight junctions. Their model is that the exocyst is needed to dock E-Cadherin-containing vesicles to the plasma membrane near tight junctions and in doing so bring PI-3K to these sites. This leads to elevated PI3P, which stimulates AKT, thereby inhibiting apoptosis. In general, I found the arguments to be believable, however some aspects need additional support.

1. In Fig 2g how does the level of AKT activity in AKT-CA cells compare to that in normal cells? The bypass of Par3 depletion-induced apoptosis by AKT-CA supports their linear model, but if the level of AKT activity is far above that of normal cells, it could be bypassing via an abnormal pathway. One could imagine that there are multiple routes contributing to apoptosis and strongly stimulating one pathway could bypass a block on a converging pathway. This would be less likely if the level of AKT activity were near normal.
2. In Fig 2k the level of the Par3R596D/K598D protein is much greater than normal. Will this allele suppress apoptosis at normal levels? Although reduced in its affinity for Pten, overproduction might compensate for the reduced affinity and restore Pten recruitment.
3. I am confused by the appearance of the E Cadherin and ATPase in Fig 3. While the images show that those proteins do not concentrate at tight junction upon Par3 depletion they do not address the nature of the structures that they go to. In panel A the E-Cadherin staining looks very different in the two cell lines. Why? What are the big blobs in the Eph4 cells? The same questions arise with the ATPase staining in panel D. Panel E shows that VSVG is blocked, but the staining looks very different from that of Cadherin and ATPase. In the case of BfA treatment in panel F the Cadherin looks like it is at the PM, just not in tight junctions. If this is the case, why would this situation lead to apoptosis when the lipid-anchored Par3 domain does not? If all of these cargoes require the exocyst for delivery to the PM, one would think that they should all accumulate in the same structures. Since the exocyst is thought to dock vesicles carrying Rab8 or Rab11 those structures should be marked with one of those Rabs. This part of the story needs more work.
4. In Fig 4 B the E Cadherin looks like it is largely at the PM in Sec8-depleted Eph4 cells. Why is this not sufficient to block apoptosis?
5. In Fig 5 A and B how much Par3 comes down with Sec8 and vice versa? The readers need to know how efficient the co-precipitation is. Does the WCL lane represent 100% input, 10%, 1%?
6. In Fig 5 K it would be better to have a control of similar size and charge as the GST-Par 3967-1045 fusion. GST is exceptionally well folded and therefore not a good control for a fusion that is highly charged and possibly not well folded.
7. Minor point: in Fig G I believe the far right lane should be labelled "+" to indicate the addition of shPar3. Otherwise it doesn't make sense.

Reviewer #3 (Remarks to the Author):

In this manuscript Ahmed and Macara provide evidence that the Par3 polarity protein interacts with exocyst components in cultured mammary epithelial cells, thereby supporting Akt-dependent survival signals. The study is very well designed and aims at obtaining mechanistic insight into how Par3 controls mammary epithelial cell survival. Moreover, mapping studies identify a critical region within Par3 that is required for interaction between Par3 and Exo70 and mediating survival signals in these cells. Interestingly, this function seems to be independent of aPKC. Using various rescue approaches the authors did considerable efforts to reveal that the reduced survival signaling downstream of Par3 is a consequence of impaired exocyst delivery to the membrane. Due to its in-depth molecular analysis and the appealing new concept that Par3 may serve as docking protein for the secretory machinery at the cortex this manuscript will in principle be of significant interest for the broad readership of Nature Communications. However, the manuscript in the current version still bears ambiguity in data interpretation, and I miss sufficient discussion regarding the relevance of the presented data for mammary homeostasis in general, also in light of the laboratory's earlier work (Refs. 8, 9, 15). Moreover, some technical issues outlined below

should be addressed.

Major points:

- The authors convincingly show that Par3 depletion in two mammary epithelial lines results in strongly increased apoptosis. In fact, apoptosis seems massive. While this part is striking, to me it remains unclear how these data can be translated to the function of Par3 in mammary morphogenesis or mammary epithelial homeostasis in vivo. On page 10/11 the authors state: "...this function is absolutely required for the survival of mammary epithelial cells.". In earlier work, however, the Macara laboratory has elegantly shown that Par3 depletion in primary mammary epithelial cells and transplantation into the fat pad does not interfere with gross formation of mammary glands and ducts, though end bud remodeling was impaired (Ref. 8). If the mechanisms the authors now uncovered in mammary epithelial cell lines were of physiologic relevance in vivo one would expect –based on the strong apoptotic phenotype presented- that mammary gland formation fails due to an inability of MECs to implant and expand in mice. How do the authors reconcile their different findings in vitro and in vivo? This should be discussed in the manuscript, next to the note that Par3 depletion in MECs results not only in apoptosis but also increased proliferation in vivo (Ref. 8). Along these lines, Par3 depletion in the mammary cell line MCF10A has also been shown to increase cell proliferation while pronounced apoptosis has not been reported in MCF10.2A, T47D and SKBR3 cells depleted for Par3 (Xue et al., NCB 2013), findings that should be dealt with in the discussion.
- The authors show that Par3 depletion in NMuMG and Eph4 cells disturbs post-Golgi trafficking, including defective E-cadherin targeting to the membrane. Earlier work by others showed that Sec6/8 form a complex with tight junction proteins and surface-labeled Ecadherin, suggesting that Sec6/8 are recruited by cell-cell contact molecules to the plasma membrane (Ref. 49). In previous seminal work the Macara laboratory reported that Par3 depletion in MDCKII cells results in impaired Tight Junction formation with defective localization of ZO-1 and the Tight Junction transmembrane protein Occludin (Ref. 9). Are these defects also seen in the mammary epithelial cell lines used, i.e. are Tight Junctions impaired in shPar3 mammary epithelial cells? In other words: Could disturbed junction architecture or composition as consequence of Par3 depletion be causal for decreased exocyst delivery to the cortex? And what is the effect of the Par3 LRD mutant on Tight Junction formation/maintenance in shPar3 NMuMG cells?
- In the last paragraph of the discussion the authors seem to aim at clarifying a putative conflict between their mammary epithelial cell in vitro data and earlier reports on increased apoptosis in mice with epidermal Par3 inactivation. However, as pointed out above, the authors have actually not explained for their mammary system how they reconcile the massive apoptosis following Par3 depletion in vitro (this study) with more mild apoptosis, additional hyperproliferation and mammary gland formation in vivo (Ref. 8). Second, their data are in fact in line with the reports in the skin epithelium, rather than being conflicting. Par3 inactivation in keratinocytes results in decreased Akt activity and activation of the intrinsic apoptosis pathway (Ref. 16). Therefore it is actually conceivable that Par3 function in exocyst-dependent survival signaling contributes not only to the survival of mammary epithelial cells but perhaps also to that of other epithelia including the epidermis. The speculation on the lethality of Par3 knockout mice thus seems exaggerated and not constructive. The future reader would instead greatly benefit from the authors' thoughts about the significance of their (mechanistically very intriguing) new data on a link between Par3 and the exocyst for breast tissue homeostasis in vivo. The authors should rephrase and focus their discussion.

Minor points:

- Quantification and statistical evaluation should be provided for Figures 6c, d, 7c, 7f.
- In several Western Blots the proteins analyzed show a different apparent molecular weight and/or differential band width in the shPar3 cells (Figure 2a,j,k, Figure 3b,g, Figure 4c). Do the authors have any explanation for this? These differences complicate a fair comparison and quantification. Providing better data or at least stating an explanation for these technical issues in the legend may help the reader judging the robustness of the phenotypes.
- Along these lines: Including data derived from primary MECs is highly appreciated as this

supports the authors' data obtained in mammary epithelial cell lines NMuMG and Eph4. However, the phenotype of decreased Akt phosphorylation in MECs is hard to see in the Western blot data (Figure 2E) as the middle band is almost double the width than that of the other lanes. As this experiment has only been performed twice, and the data shown appear to be the best example, it would be important to further corroborate these findings by repeat experiments, including statistical assessment (n at least 3).

- Is the extent of apoptosis in shPar3 MECs comparable to that of shPar3 NMuMG and Eph4 cells?
- Figure 5b: Please specify in the legend and figure what '- Sec8 IP' exactly refers to. Did the authors control for potential unspecific binding of Sec8 to beads or IgGs? Similarly, please specify what 'mock' refers to in Figure 5a.
- Figure 2d-f: Quantification of pAkt data in the mammary epithelial cell line shown in d should follow in panel e, not f, to clearly connect them to NMuMG cells.
- It is slightly confusing that the authors chose Sec6 for costaining with Par3 in immunofluorescence, Sec8 for biochemical interaction studies, and eventually depict Sec3 as Par3 interacting protein in the final model. While this may in part have technical reasons it would help to clarify the choices made somewhere in the legend or manuscript.

Point by point response to reviewer comments

Reviewer 1.

- 1. The title of the paper sounds as it reports two points; PAR3 is an exocyst receptor and the receptor is essential for mammary cell survival. However, this reviewer strongly feels that the latter is not fully supported by experimental evidence and that the title might be misleading. Thus, I suggest that the title should be changed.*

We believe that we have now added convincing data in support of the conclusion that PAR3 is required for mammary cell survival. We show that in primary luminal mammary cells, freshly isolated by FACS from murine mammary glands, silencing of Par3 results in decreased phospho-Akt, and increased caspase 3 cleavage, as occurs in the mammary epithelial cell lines, followed by cell death (new Figure 2).

- 2. Previous studies on a variety of species such as Drosophila and mammals suggested that PAR3 is involved in localization of apical proteins. What does the central conclusion of the present manuscript mean in the light of the role of PAR3 on localization of apical proteins.*

PAR3 has multiple functions, including the localization of aPKC to the apical surface. Apical aPKC can then exclude basolateral proteins from associating with the apical domain. We do not have evidence to show that the exocyst is involved in aPKC delivery, or that the exocyst interacts with aPKC in epithelial cells, so exocyst function might not directly impact exclusion of apical proteins. Additionally, we note that although in *Drosophila* PAR3 is necessary for the apical localization of Crumbs (at least in some epithelial tissues), this does not appear to be the case in mammalian epithelial cells. We now discuss this point in the revised manuscript.

- 3. Does NMuMG and Eph4 cells show polarized phenotype? Do they have distinct lateral and apical domains? Do they have tight junctions?*

It is important to note that neither NMuMG cells nor Eph4 cells are cancer cell lines. They are both immortalized but otherwise “normal” luminal mammary epithelial cells. (NMuMG = Normal murine Mammary Gland). The apical and lateral domains are properly separated in both NMuMG and Eph4 cells, and both cells form tight junctions, as has been reported in previous publications (Miettinen et al., 1994; Nagaoka et al., 2012; Blackman et al., 2005). However, per the reviewer’s suggestion, we have now added new data (Supplementary figure 1) showing the apical localization of ZO-1 and YFP-Par3, and lateral localization of the E-Cadherin binding partner, β -catenin, in both NMuMG and Eph4 cells. ZO-1 and YFP-Par3 colocalize at the apical region of the cells, but their abundance sharply drops in plasma membrane regions where β -catenin localization is concentrated. (We used β -catenin as a surrogate marker for E-Cadherin because our ZO-1 and E-Cadherin antibodies are both mouse monoclonal antibodies).

- 4. Does PAR3 depletion affect localization of E-cad and Na⁺-K⁺-ATPase (NKA) as cancer cell lines?*

Low passage normal murine mammary epithelial cells (NMuMG) cells are non-transformed cells, and maintain epithelial characteristics. Eph4 is also a non-tumorigenic cell line derived from spontaneously immortalized mouse mammary gland epithelial cells, and is highly polarized. They have both been widely used as models for polarized epithelial cells (Miettinen

et al., 1994; Nagaoka et al., 2012; Blackman et al., 2005). We have now added new data showing that FACS-purified primary luminal epithelial cells freshly isolated from mice behave in the same way as these cell lines in response to depletion of Par3.

5. *The only data for primary mammary epithelial cells is the one shown in Fig2e. However, the data is not clear; some bands are deformed suggesting sample overloading, casting doubts for the authors conclusion.*

We agree with the reviewer's concern and repeated the experiment using FACS-purified primary luminal mammary epithelial cells freshly isolated from mice. The newly generated data are provided in Figs 2f-i. We also provide statistics from three independent experiments, shown in Fig 2i.

One technical issue is that the primary cells die quickly upon depletion of Par3, so recovery of sufficient cells for subsequent assays precludes extensive washing of the cells to remove serum albumin, which can lead to band distortion during the gel electrophoresis.

6. *There is no description how the M/C ratio of PH-AKT-GFP is quantified from the 2D staining data in Fig2i.*

We appreciate the reviewer catching this oversight. We have now added the description in the methods section, "measurement of PIP3 levels", explaining the quantification.

7. *The authors use the word "survival or apoptosis" based on the appearance of bands of cleaved caspase-3 in most cases. This is misleading and they should precisely describe their data. (Texts for Fig 2g, Fig2j, and Fig2k; Fig4e)*

The text has been amended as suggested by the reviewer.

8. *Does the LRD sequence motif conserved for C. elegans PAR3 (Fig5d)?*

We have compared the PARD3 sequence to the *C. elegans* homolog and now report it in Fig 7d. The level of similarity is much lower than for other species. Nonetheless, Jeremy Nance has reported links between Pard3 and the exocyst in *C. elegans* that are highly consistent with our conclusions, as we mentioned in the discussion.

Reviewer 2.

1. In Fig 2g how does the level of AKT activity in AKT-CA cells compare to that in normal cells? The bypass of Par3 depletion-induced apoptosis by AKT-CA supports their linear model, but if the level of AKT activity is far above that of normal cells, it could be bypassing via an abnormal pathway. One could imagine that there are multiple routes contributing to apoptosis and strongly stimulating one pathway could bypass a block on a converging pathway. This would be less likely if the level of AKT activity were near normal.

We agree that the constitutively active mutant of AKT has higher activity than the AKT in normal cells. However, while the reviewer raises a good point here, this concern was addressed in our original experiments where, instead of over-expressing AKT-CA, we knocked down Pten – a negative regulator - to rescue endogenous Akt activity (shown in Fig 3a)

2. In Fig 2k the level of the Par3R596D/K598D protein is much greater than normal. Will this allele suppress apoptosis at normal levels? Although reduced in its affinity for Pten,

overproduction might compensate for the reduced affinity and restore Pten recruitment.

As shown in supplementary figure 4g, we tried to assess the interaction between exogenously expressed HA-tagged PTEN and overexpressed PAR3 wild-type or the R596D/K598D mutant. Even with substantial over-expression we could not see any significant interaction between PTEN and WT PAR3. Therefore, we have no reason to expect that high levels of the mutant could compensate.

3. I am confused by the appearance of the E Cadherin and ATPase in Fig 3. While the images show that those proteins do not concentrate at tight junction upon Par3 depletion they do not address the nature of the structures that they go to. In panel A the E-Cadherin staining looks very different in the two cell lines. Why? What are the big blobs in the Eph4 cells? The same questions arise with the ATPase staining in panel D. Panel E shows that VSVG is blocked, but the staining looks very different from that of Cadherin and ATPase. In the case of BFA treatment in panel F the Cadherin looks like it is at the PM, just not in tight junctions. If this is the case, why would this situation lead to apoptosis when the lipid-anchored Par3 domain does not? If all of these cargoes require the exocyst for delivery to the PM, one would think that they should all accumulate in the same structures. Since the exocyst is thought to dock vesicles carrying Rab8 or Rab11 those structures should be marked with one of those Rabs. This part of the story needs more work.

First, we agree with the reviewer that the response of the two cell lines to silencing of Par3 is somewhat different. We believe that in part this is because NMuMG cells have less robust intercellular junctions than Eph4 cells – E-cadherin is internalized faster, and even at steady state in WT cells there is more E-cadherin in cytoplasmic vesicles than is present in Eph4 cells. In addition, as suggested by the reviewer, we have addressed this point by examining the co-localization of E-cadherin with Rab11 and Rab8. Because there are no reliable antibodies against these proteins (which we tested extensively), we expressed YFP-tagged Rab11 and Rab8 to assess co-localization with the internalized E-cadherin. We have added a new Figure 5, which shows substantial colocalization of Rab11 (new Fig 5a) with internalized E-cadherin after Par3 depletion in both Eph4 and NMuMG cells.

4. In Fig 4 B the E Cadherin looks like it is largely at the PM in Sec8-depleted Eph4 cells. Why is this not sufficient to block apoptosis?

Please note that this figure is now Figure 6B. We captured the images for the Eph4 and NMuMG cells at the same time after transduction; but because Eph4 cells have much more robust junctions the rate at which they dissipate is slower, and the accumulation of the E-cadherin in cytoplasmic vesicles is also much slower. Nonetheless, most of the Eph4 cells eventually died, 1 – 2 days after apoptosis of the NMuMG cells.

5. In Fig 5 A and B how much Par3 comes down with Sec8 and vice versa? The readers need to know how efficient the co-precipitation is. Does the WCL lane represent 100% input, 10%, 1%?

This is now indicated in the new figures, which are now Fig 7a and b.

6. In Fig 5 K it would be better to have a control of similar size and charge as the GST-Par 3967-1045 fusion. GST is exceptionally well folded and therefore not a good control for a fusion that is highly charged and possibly not well folded.

This is a good point. We have now added a separate blot where we used GST-MARCKs 151-175 (which also has a polybasic region) as a control, and which is closer in size to GST-PAR3b 967-1045 than GST alone. Little to no binding of the GST-MARCKS was detected to exocyst.

Minor point: in Fig 1G I believe the far right lane should be labelled “+” to indicate the addition of shPar3. Otherwise it doesn’t make sense.

We appreciate the reviewer catching this oversight and this has now been corrected.

Reviewer 3.

1. The authors convincingly show that Par3 depletion in two mammary epithelial lines results in strongly increased apoptosis. In fact, apoptosis seems massive. While this part is striking, to me it remains unclear how these data can be translated to the function of Par3 in mammary morphogenesis or mammary epithelial homeostasis in vivo. On page 10/11 the authors state: “...this function is absolutely required for the survival of mammary epithelial cells.”. In earlier work, however, the Macara laboratory has elegantly shown that Par3 depletion in primary mammary epithelial cells and transplantation into the fat pad does not interfere with gross formation of mammary glands and ducts, though end bud remodeling was impaired (Ref. 8). If the mechanisms the authors now uncovered in mammary epithelial cell lines were of physiologic relevance in vivo one would expect –based on the strong apoptotic phenotype presented- that mammary gland formation fails due to an inability of MECs to implant and expand in mice. How do the authors reconcile their different findings in vitro and in vivo? This should be discussed in the manuscript, next to the note that Par3 depletion in MECs results not only in apoptosis but also increased proliferation in vivo (Ref. 8). Along these lines, Par3 depletion in the mammary cell line MCF10A has also been shown to increase cell proliferation while pronounced apoptosis has not been reported in MCF10.2A, T47D and SKBR3 cells depleted for Par3 (Xue et al., NCB 2013), findings that should be dealt with in the discussion.

We thank the reviewer for these comments. However, we note that loss of Par3 did severely disrupt mammary gland formation (McCaffrey et al., 2009), resulting in a dramatic decrease in the outgrowth of the ductal tree into the fat pad, plus branching defects, a profound disorganization of the bilayered structure of the ducts, loss of normal lineage specification, and increased apoptosis and proliferation. However, it is true that the level of apoptosis observed in the regenerated glands was lower than what we observe in vitro – possibly because of the impact of neighboring myoepithelial cells or stromal cells in the tissue, or systemic factors that suppress apoptosis. We now discuss this in the manuscript.

We also demonstrated that the expression of an oncogene (NICD) suppresses apoptosis in response to loss of Par3 (Archibald et al, 2014). The cell lines T47D and SKBR3 cells, referred to by the reviewer, are transformed, and therefore very likely also have the capability to evade apoptosis. Additionally, the MCF10A cells used by Xue et al expressed activated ErbB2, which stimulates AKT phosphorylation and will counteract the apoptotic signal that would otherwise be triggered by loss of Par3. As we discussed in the original manuscript, knockout of Par3 in the epidermis of mice does lead to increased apoptosis, just as we observed in the mammary gland and in our mammary epithelial cell lines.

To validate the biological significance of our data, we have now FACS-purified primary luminal epithelial cells from murine mammary glands, and show that in these freshly isolated cells the depletion of Par3 results in decreased Akt phosphorylation and increased apoptosis (new panels in Figure 2).

2. The authors show that Par3 depletion in NMuMG and Eph4 cells disturbs post-Golgi trafficking, including defective E-cadherin targeting to the membrane. Earlier work by others showed that Sec6/8 form a complex with tight junction proteins and surface-labeled Ecadherin, suggesting that Sec6/8 are recruited by cell-cell contact molecules to the plasma membrane (Ref. 49). In previous seminal work the Macara laboratory reported that Par3 depletion in MDCKII cells results in impaired Tight Junction formation with defective localization of ZO-1 and the Tight Junction transmembrane protein Occludin (Ref. 9). Are these defects also seen in the mammary epithelial cell lines used, i.e. are Tight Junctions impaired in shPar3 mammary epithelial cells? In other words: Could disturbed junction architecture or composition as consequence of Par3 depletion be causal for decreased exocyst delivery to the cortex? And what is the effect of the Par3 LRD mutant on Tight Junction formation/maintenance in shPar3 NMuMG cells?

These are good points. We have now directly addressed them in new experiments, shown in Figure 9h. Tight junctions are lost upon depletion of Par3, and mutants of Par3 lacking the exocyst binding region (EBR) could not rescue tight junction formation. This is not too surprising since adherence junction (AJ) formation usually is a prerequisite for TJ formation, and AJs do not form correctly in the absence of Par3 in mammary cells. Moreover, our data are consistent with previous results (Chen & Macara, 2005) where we showed that expression of the c-terminus of PAR3c (aa620-1266; which included PDZ3 and the exocyst binding region but not the dimerization, PDZ1, PDZ2 and aPKC binding region) was sufficient to rescue tight junction defects caused by loss of Par3 in MDCK cells. We have now included this in our discussion. We agree that loss of junctions might be partly responsible for decreased exocyst delivery, but we believe that our identification of direct binding between Par3 and Exo70, plus the inability of the EBR mutants of Par3 to rescue E-cadherin localization and cell survival together provide a strong argument that exocyst binding to Par3 is important for exocyst function.

3. In the last paragraph of the discussion the authors seem to aim at clarifying a putative conflict between their mammary epithelial cell in vitro data and earlier reports on increased apoptosis in mice with epidermal Par3 inactivation. However, as pointed out above, the authors have actually not explained for their mammary system how they reconcile the massive apoptosis following Par3 depletion in vitro (this study) with more mild apoptosis, additional hyperproliferation and mammary gland formation in vivo (Ref. 8). Second, their data are in fact in line with the reports in the skin epithelium, rather than being conflicting. Par3 inactivation in keratinocytes results in decreased Akt activity and activation of the intrinsic apoptosis pathway (Ref. 16). Therefore, it is actually conceivable that Par3 function in exocyst-dependent survival signaling contributes not only to the survival of mammary epithelial cells but perhaps also to that of other epithelia including the epidermis. The speculation on the lethality of Par3 knockout mice thus seems exaggerated and not constructive. The future reader would instead greatly benefit from the authors' thoughts about the significance of their (mechanistically very intriguing) new data on a link between Par3 and the exocyst for breast tissue homeostasis in vivo. The authors should rephrase and focus their discussion.

We agree that our data are entirely consistent with the effects of Par3 knockout in the murine epidermis, and mention this in the discussion. In the mammary gland in vivo we reported that silencing of Par3 reduced outgrowth by 80 - 90%, despite increased staining by the proliferation marker Ki67 (McCaffrey 2009). Since we also observed increased apoptosis (cleaved caspase 3 staining) we concluded that the pronounced decrease in outgrowth was because of apoptosis. We note that follow-up studies, using primary murine mammospheres, are fully consistent with

this conclusion: mammosphere growth was compromised by silencing of Par3, and apoptosis was increased (Archibald et al, 2014, Figure 1e,f). Caspase inhibition resulted in a dramatic increase in mammosphere growth (Archibald, Figure j, k). We have now further corroborated these results using FACS-purified primary luminal epithelial cells from mouse mammary glands, which also show substantial apoptosis (and reduced phospho-AKT) in response to silencing of Par3 (new Figure 2). We now discuss these points in the revised manuscript.

4. Quantification and statistical evaluation should be provided for Figures 6c, 6d, 7c, 7f.

Quantifications have been added as requested.

5. In several Western Blots the proteins analyzed show a different apparent molecular weight and/or differential band width in the shPar3 cells (Figure 2a,j,k, Figure 3b,g, Figure 4c). Do the authors have any explanation for this? These differences complicate a fair comparison and quantification. Providing better data or at least stating an explanation for these technical issues in the legend may help the reader judging the robustness of the phenotypes.

As the cells were dying we had to collect the cells in a manner that left some BSA from the culture media, which caused the contaminating protein to interfere with gel migration (BSA and AKT are both around 66kDa). We now mention this technical problem in the results section.

6. Along these lines: Including data derived from primary MECs is highly appreciated as this supports the authors' data obtained in mammary epithelial cell lines NMuMG and Eph4. However, the phenotype of decreased Akt phosphorylation in MECs is hard to see in the Western blot data (Figure 2E) as the middle band is almost double the width than that of the other lanes. As this experiment has only been performed once, and the data shown appear to be the best example, it would be important to further corroborate these findings by repeat experiments, including statistical assessment (n at least 3).

We agree with the reviewer, and have repeated these experiments, using FACS-purified primary luminal epithelial cells freshly prepared from mouse mammary glands. Revised western blots and quantification now appears in the new Fig 2f-i.

7. Is the extent of apoptosis in shPar3 MECs comparable to that of shPar3 NMuMG and Eph4 cells?

Although the phenotype is less severe, it is still significant as shown in the new Fig 2h and 2i. It is important to note that the MECs used in this original experiment consisted of a mixture of basal and luminal cells. So far as we know, the basal cells are resistant to silencing of Par3 and this might account in part for the less severe phenotype both in vitro and in vivo.

8. Figure 5b: Please specify in the legend and figure what '- Sec8 IP' exactly refers to. Did the authors control for potential unspecific binding of Sec8 to beads or IgGs? Similarly, please specify what 'mock' refers to in Figure 5a.

This has now been corrected and appears as Fig 7b. The control IP was done with mouse IgG to match the Sec8 mouse antibody.

9. Figure 2d-f: Quantification of pAkt data in the mammary epithelial cell line shown in d should follow in panel e, not f, to clearly connect them to NMuMG cells.

We have changed this as suggested.

10. It is slightly confusing that the authors chose Sec6 for costaining with Par3 in immunofluorescence, Sec8 for biochemical interaction studies, and eventually depict Sec3 as Par3 interacting protein in the final model. While this may in part have technical reasons it would help to clarify the choices made somewhere in the legend or manuscript.

We stained for both Sec8 and Sec6 to provide validation by using antibodies against 2 subunits of the same complex. Limitations in available antibodies precluded high quality immunostaining for Exo70 or Sec3. We did not intend to show Sec3 as a Par3-binding protein in the model, and this has now been modified.

REVIEWERS' COMMENTS:

Reviewer #1 (Remarks to the Author):

This reviewer is satisfied with the revised manuscript and think that this manuscript is now suitable for publication.

Reviewer #2 (Remarks to the Author):

The revised manuscript has addressed most of my prior concerns. Just two points:

1. In Fig 7B there seems to be a labeling error, both the top panel and the middle panel are labeled "WB Par3 endogenous", but I am pretty sure based on the position of the band that the middle panel should be WB Sec8 endogenous.
2. There are strong parallels between the interaction of Exo70 with Par3 and a published interaction of yeast Exo70 with the polarity determinant scaffold protein Bem1. These parallels should be discussed and the sequences of the Exo70 binding domains in Par3 and Bem1 should be compared to see if there are any notable similarities.

Reviewer #3 (Remarks to the Author):

The authors have perfectly addressed most of my comments. Particularly the additional primary mammary cell data significantly improved the manuscript. However, there are still a few remaining points that require attention.

- Original comment by this reviewer (8):

"Figure 5b: Please specify in the legend and figure what '- Sec8 IP' exactly refers to. Did the authors control for potential unspecific binding of Sec8 to beads or IgGs? Similarly, please specify what 'mock' refers to in Figure 5a.."

authors' response: "This has now been corrected and appears as Fig 7b. The control IP was done with mouse IgG to match the Sec8 mouse antibody."

reviewer: Fig 7b has been clarified. There is still inconsistency in revised Fig7a though. The legend now says: "(a) Co-immunoprecipitation of Myc-PAR3 and Sec8. HEK293T cells were transfected with Myc-PAR3b and immunoprecipitated with anti-Myc (clone 9E10) antibody and immunoblotted with anti-Myc-HRP and anti-Sec8 antibodies."

What is the control shown in left lane? Were both samples transfected with Myc-PAR3, with subsequent IPs for myc and mouse control IgGs? Or was the left sample transfected with an empty vector or else (original figure stated "mock" without further detail), and then both lysates (mock and Myc-Par3-transfected) were subjected to IP using anti-myc antibodies (as suggested by the label on the right) – please correct and specify for the reader.

- Page 7 of revised manuscript: Figure citations to figure 3d-f seem to be incorrect, please adapt to revised figures.
- Page 7: "Together, these data support a model in which Par3 is absolutely required for the delivery of lateral membrane proteins from the Golgi, ...". To this reviewer the term "absolutely" seems overstated considering the experimental evidence provided.
- Page 8: "Silencing of Sec8 caused E-Cadherin mislocalization in both NMuMG and Eph4 cells (Figure 6b).": the label in that figure states "shPar3". I assume the authors meant "shSec8".

- Suppl. Fig 5c is not cited in the manuscript text, and the hPAR3 mutant information in the legend should be checked (should read R596D,K598D).

Point by point response to reviewer comments

Reviewer 2.

1. In Fig 7B there seems to be a labeling error, both the top panel and the middle panel are labeled “WB Par3 endogenous”, but I am pretty sure based on the position of the band that the middle panel should be WB Sec8 endogenous.

The reviewer is correct, and the mistake has been rectified.

2. There are strong parallels between the interaction of Exo70 with Par3 and a published interaction of yeast Exo70 with the polarity determinant scaffold protein Bem1. These parallels should be discussed and the sequences of the Exo70 binding domains in Par3 and Bem1 should be compared to see if there are any notable similarities.

This is a very interesting point raised by the reviewer. We looked for sequence similarity between the Par3-LRD and the Bem1 exocyst binding region (aa309-470). However, there was no significant similarity, as compared to a random sequence picked from another region of Par3.

Reviewer 3.

1. Original comment by this reviewer (8):
“Figure 5b: Please specify in the legend and figure what ‘- Sec8 IP’ exactly refers to. Did the authors control for potential unspecific binding of Sec8 to beads or IgGs? **Similarly, please specify what ‘mock’ refers to in Figure 5a..**”

authors’ response: “This has now been corrected and appears as Fig 7b. The control IP was done with mouse IgG to match the Sec8 mouse antibody.”

reviewer: Fig 7b has been clarified. There is still inconsistency in revised Fig7a though. The legend now says: “(a) Co-immunoprecipitation of Myc-PAR3 and Sec8. HEK293T cells were transfected with Myc-PAR3b and immunoprecipitated with anti-Myc (clone 9E10) antibody and immunoblotted with anti-Myc-HRP and anti-Sec8 antibodies.”

What is the control shown in left lane? Were both samples transfected with Myc-PAR3, with subsequent IPs for myc and mouse control IgGs? Or was the left sample transfected with an empty vector or else (original figure stated “mock” without further detail), and then both lysates (mock and Myc-Par3-transfected) were subjected to IP using anti-myc antibodies (as suggested by the label on the right) – please correct and specify for the reader.

This is now clarified in the figure legend of Fig 7a – cells were transfected with pKmyc-PAR3 or with pKVenus (as a negative control) and lysates were immunoprecipitated with anti-Myc antibody. We have corrected the labeling of the figure accordingly.

2. Page 7 of revised manuscript: Figure citations to figure 3d-f seem to be incorrect, please adapt to revised figures.

We appreciate the reviewer noticing this oversight, and the mistake has been corrected in the revised manuscript.

3. Page 7: “Together, these data support a model in which Par3 is absolutely required for the delivery of lateral membrane proteins from the Golgi, ...”. To this reviewer the term “absolutely” seems overstated considering the experimental evidence provided.

We agree and have removed the word “absolutely” from the sentence as per the reviewer’s comment. This sentence now appears at the top of Page 8.

4. Page 8: “Silencing of Sec8 caused E-Cadherin mislocalization in both NMuMG and Eph4 cells (Figure 6b).”: the label in that figure states “shPar3”. I assume the authors meant “shSec8”.

We have now corrected the figure label to “shSec8”, which is the correct label.

5. Suppl. Fig 5c is not cited in the manuscript text, and the hPAR3 mutant information in the legend should be checked (should read R596D, K598D).

We appreciate the reviewer bringing this to our attention. We have now added a sentence in Page 7 to cite this figure.

“Furthermore, it is unlikely that the phenotype is caused by disruption of a Par3-Pten interaction, as expression of PAR3(R596D, K598D) was able to rescue E-Cadherin localization to the plasma membrane (Supplementary Fig. 5c).”

We have also corrected the label in the figure and the figure legend.